# Evidence that dengue virus envelope protein in patient platelets originates from infection of megakaryocytic precursors

Marcos Alvarado Silva[1,2], Tannya Karen Castro Jimenez[1], Edwin Antonio Lopez Kelly[1], Nallely Diaz Lima[3], José Alberto San Juan Luis[3], Diego Sait Cruz Hernández[2], Sergio Roberto Aguilar Ruiz[2], José Bustos Arriaga[1*]

1 Laboratorio de Biología Molecular e Inmunología de arbovirus, Unidad de Biomedicina, Facultad de Estudios Superiores Iztacala, Universidad Nacional Autónoma de México, Tlalnepantla, México, 2 Departamento de Biomedicina Experimental, Facultad de Medicina y Cirugía de la Universidad Autónoma 'Benito Juárez' de Oaxaca, Oaxaca, México, 3 OaxacaLab Laboratorio de análisis Clínicos, Oaxaca, México

* jose.bustos@unam.mx

## Abstract

Dengue virus infection is a major global health problem with clinical outcomes ranging from mild febrile illness to severe disease. Although a reduction in platelet counts is a common feature of dengue, the mechanisms underlying the presence of viral antigens in circulating platelets remain incompletely understood. In this study, we investigated whether infection of megakaryocytic precursors contributes to the generation of platelets carrying dengue virus proteins. Using the human megakaryocytic precursor cell lines K562 and MEG-01 cell lines as in vitro models, we show that reference strains and clinical isolates of dengue virus infect megakaryocytic precursors and modulate their differentiation, as evidenced by increased expression of the differentiation marker CD41 and the production of platelet-like particles. Flow cytometry analysis demonstrated that PLPs released from infected precursors contained precursor-derived PLP-associated dengue virus envelope protein. Correlation analyses revealed significant associations between the extent of precursor infection and both differentiation markers and PLP-associated E protein levels at early time points. Consistent with these findings, dengue virus E protein was detected in platelets from dengue patients, while no statistically significant differences were observed between clinical severity groups, a trend toward higher proportions of E+ platelets were observed in patients with dengue with warning signs or severe dengue. Spearman correlation analyses further supported an association between in vitro precursor infection and the generation of E-positive PLPs. In contrast, platelets from healthy donors incubated ex vivo with dengue virus did not acquire E protein, indicating that direct uptake or infection of circulating platelets is inefficient. Together, these results support a model in which infection of megakaryocytic precursors contributes to the generation of platelets carrying dengue virus antigens, suggesting that

**Data availability statement:** All data underlying the findings reported in this study are provided within the manuscript. Raw numerical data used to generate all figures and statistical analyses are available as Supporting Information.

**Funding:** This work was supported by the following sources: 1.Universidad Nacional Autónoma de México (UNAM), PAPIIT-DGAPA Grant numbers: IN200821 and IN215224 Awarded to: JBA Funder website: https://dgapa.unam.mx/index.php/impulso-a-la-investigacion/papiit 2. Secretaría de Ciencia, Humanidades, Tecnología e Innovación (Secihti, Mexico), Ciencia de Frontera 2023 program Grant number: CF-2023-I-478 Awarded to: JBA Funder website: https://secihti.mx/ciencia-de-frontera/ 3. Secretaría de Ciencia, Humanidades, Tecnología e Innovación (Secihti, Mexico), National Graduate Scholarship Program Doctoral scholarship awarded to: MAS Funder website: https://secihti.mx/becas_posgrados/becas-nacionales/becas-nacionales-de-posgrado/ 4. Consejo Mexiquense de Ciencia y Tecnología (COMECYT), Program Investigación de Mujeres Científicas 2022 Grant number: FICDTEM-2023-21 Awarded to: TKCJ Funder website: https://comecyt.edomex.gob.mx/mujeres-cientificas The funders had no role in study design, data collection and analysis, decision to publish, or preparation of the manuscript.

**Competing interests:** The authors have declared that no competing interests exist.

platelet-associated E protein may reflect bone marrow involvement during dengue infection.

## Introduction

Dengue virus *(DENV; species Orthoflavivirus denguei* according to ICTV classification). It is an enveloped virus with a single positive-sense RNA genome containing one open reading frame that is translated as a polyprotein and processed into three structural and seven non-structural proteins by the viral protease NS2/NS3 and host proteases [1].

Four DENV serotypes have been identified to date, transmitted to humans mainly by *Aedes aegypti* and *Aedes albopictus* mosquitoes. Infection with any of the four serotypes can result in dengue disease, classified by the World Health Organization into dengue without warning signs (DWW), dengue with warning signs (DWS), and severe dengue (SD), the latter characterized by plasma leakage, bleeding, and organ involvement. Thrombocytopenia is one of the most frequent hematological abnormalities in DWS and SD, affecting up to 70% of patients [2–5].

Platelets are generated by megakaryocytes through the process of megakaryopoiesis. This process involves the differentiation of hematopoietic stem cells into megakaryocytic–erythroid progenitors (MEPs), which then mature under the influence of cytokines such as thrombopoietin and interleukins into megakaryoblasts and ultimately megakaryocytes. During differentiation, markers such as CD41 and CD42 are upregulated, and pro-platelet structures are formed, which fragment into circulating platelets [6–8].

Previous studies have demonstrated that DENV RNA can be detected in immature bone marrow progenitors and that MEP cell lines are permissive to direct DENV infection. These observations suggest that infection of precursor cells may contribute to platelet-associated viral antigen detection independently of immune complex–mediated mechanisms. Building on these observations, we hypothesized that dengue virus infection of megakaryocytic precursors not only alters their differentiation but also facilitates the transfer of viral components, particularly the envelope (E) protein, to the platelets they generate. The objective of this study was to evaluate the differentiation capacity of MEP models infected with clinical DENV isolates and to determine whether the precursor infection is associated with the generation of platelet-associated dengue E protein [9–11].

## Materials and methods

### Patient samples

Eleven blood samples were collected from donors in Oaxaca City during the 2022 dengue outbreak. Dengue virus infection was confirmed by rapid immunochromatography assays detecting NS1 antigen, and by serological testing for DENV and ZIKV IgM and IgG antibodies. All samples were additionally analyzed by complete blood count and hepatic enzyme measurements.

## Ethics statement

Clinical isolates used in this study were obtained between 01/01/2016 and 31/12/2021 as part of previously approved projects at the Facultad de Estudios Superiores Iztacala, Universidad Nacional Autónoma de México (UNAM). The study protocols were reviewed and approved by the Institutional Ethics Committees (approval numbers CE/FESI/112017/1224 and CE/FESI/062018/1257). All participants, or their legal guardians when applicable, provided written informed consent at the time of sample collection. For the present work, clinical isolates were accessed for research purposes between 01/01/2020 and 31/12/2025. Samples were anonymized prior to analysis, and the authors had no access to personally identifiable patient information. All procedures were conducted in accordance with institutional guidelines and the Declaration of Helsinki. The biometric and clinical information obtained from patient samples is summarized in Table 1.

## Cell lines

We used two cellular models from different stages in the megakaryopoiesis; MEG-01 cell line (ATCC CRL-2021) was used as a model of megakaryoblast and K562 cells (ATCC CCL-243) as a model of the megakaryocyte-erythroid progenitor stage [12,13]. All infections were performed in the absence of dengue-immune serum, and antibody-dependent enhancement was not evaluated in this study. Vero cell line (ATCC CCL-81) was used for virus stock preparations and titration as previously described [14]. MEG-01 and K562 cells were cultured in Roswell Park Memorial Institute medium (RPMI) (Biowest) and supplemented with 10% of fetal bovine serum (FBS, Biowest) and 1% of an antibiotic/antimycotic cocktail (Bowest). Vero cells were cultured in RPMI supplemented with 5% of FBS and an antibiotic/antimycotic cocktail. The cells were maintained at 37°C in a humidified atmosphere with 5% $CO_2$.

## Virus isolates

Cells were infected with DENV1 and Zika virus (ZIKV) reference strains (Table 2), as well as with DENV2 clinical isolates obtained by our research group from patients with dengue disease presenting different degrees of thrombocytopenia. A ZIKV isolate previously obtained in Oaxaca (ZIKV-Oax) was also included. Platelet counts at the time of sampling are provided in Table 3 to contextualize thrombocytopenia severity. (Table 3). Isolates were derived from blood samples collected during the febrile stage. Diluted sera were inoculated onto Vero cell monolayers, and multiple blind passages were performed until viral isolation was confirmed. Only isolates with fewer than ten passages were used. Viral stocks were quantified by focus-forming assay, and dengue serotype was determined using the nested RT-PCR protocol of Lanciotti et al [15]. Information on thrombocytopenia in the source patients and the serotype of each isolate is listed in Table 3.

MEG-01 cells were used to evaluate differentiation-associated effects of DENV infection using well-characterized reference strains (DENV1 and DENV4), whereas comparative analyses involving multiple clinical isolates were performed in K562 cells due to their higher permissivity and experimental reproducibility. For infection assays, MEG-01 cells were infected at a multiplicity of infection (MOI) of 0.1, while K562 cells were infected at an MOI of 1, based on preliminary titration assays and previously reported permissivity of these cell lines. All clinical isolates were normalized and used at the same MOI in K562 cells. Viral adsorption was performed for 1 h at 37°C with gentle agitation, followed by removal of unbound virus and incubation in fresh complete RPMI medium.

Platelet counts correspond to the clinical laboratory measurement obtained at the time of sample collection. All viral isolates were compared using equivalent multiplicities of infection. Although isolate-dependent differences in replication kinetics cannot be excluded, normalization by infectious units allowed comparative analysis across experiments.

## Virus propagation and storage

Virus seeds were used to infect confluent Vero cell monolayers for 1 h. After infection, cells were washed with phosphate-buffered saline (PBS) to remove unbound virus, and fresh culture medium was added to facilitate propagation.

**Table 1. Clinical and laboratory characteristics of dengue patients included in the study.**

| | Patient 1 | Patient 2 | Patient 3 | Patient 4 | Patient 5 | Patient 6 | Patient 7 | Patient 8 | Patient 9 | Patient 10 | Patient 11 |
|---|---|---|---|---|---|---|---|---|---|---|---|
| **Sex** | Male | Male | Male | Male | Female | Female | Female | Male | Female | Male | Female |
| **Age (years)** | 11 | 3 | 21 | 42 | 43 | 11 | 54 | 70 | 41 | 17 | 48 |
| **DENV NS1** | + | – | + | + | + | – | + | + | + | + | – |
| **DENV IgM** | + | + | – | – | – | + | – | – | – | – | + |
| **DENV IgG** | – | + | + | – | – | + | – | – | – | – | – |
| **Hgb (g/dL)** | 15.8 | 12.2 | 16.3 | 17 | 13.4 | 15.3 | **12.6** | 15.5 | 14.3 | **17.3** | 12.8 |
| **Hct (%)** | **44.7** | **35** | 49 | 52.1 | 39.7 | **48.5** | **38.3** | 42.9 | 42.2 | **51.4** | **38** |
| **RBC (x10$^{12}$/L)** | 5.23 | 4.13 | 5.25 | 5.29 | **4.25** | **5.42** | **4.08** | 4.66 | 4.51 | **5.9** | **4.19** |
| **RDW (%)** | 13.3 | 13.4 | 13.1 | 12.7 | 12.7 | 13.1 | 13.9 | 14 | 14 | 13.1 | **14.1** |
| **MCV (fL)** | 85.3 | 84.7 | 93.3 | 98.4 | 93.4 | 89.5 | 93.9 | 92.1 | 93.6 | 87.1 | 90.7 |
| **MCH (pg)** | 30.1 | 29.6 | 31.1 | **32.1** | 31.5 | 28.2 | 30.9 | **33.2** | 31.6 | 28.2 | 30.5 |
| **MCHC (g/dL)** | 34.6 | **34.6** | 32.5 | **30.6** | 33 | 32.2 | 32.2 | 34.2 | 32.4 | 32.9 | 32.5 |
| **PLT (x10$^9$/L)** | **81** | 198 | **113** | 194 | **64** | 242 | 220 | **60** | **101** | **137** | 238 |
| **MPT (fL)** | 8.5 | 7.7 | 9.4 | 9.5 | **11.1** | 7.9 | 9.1 | **11.4** | 9.7 | 9.6 | **7.2** |
| **WBC (x10$^9$/L)** | **2.1** | 5.07 | **3.2** | 8.38 | **2.74** | 10.52 | 8.06 | 5.42 | **1.88** | 5.47 | **2.98** |
| **Neutrophils (%)** | 56 | **41** | 56 | 69 | **83** | **74** | **77** | 65 | 69 | **72** | 62 |
| **Eosinophils (%)** | 0 | 0 | 1 | **5** | 1 | 0 | 1 | 1 | 1 | 1 | 0 |
| **Basophils (%)** | 0 | 0 | 1 | 1 | 1 | 0 | 0 | 0 | 0 | 0 | 0 |
| **Monocytes (%)** | 5 | 7 | 5 | 8 | 5 | 4 | 5 | 6 | 5 | 7 | 7 |
| **Lymphocytes (%)** | 39 | **52** | 37 | **17** | **10** | 22 | **17** | 28 | 25 | 20 | 31 |
| **Bands (%)** | 0 | 0 | 0 | 0 | 0 | 0 | **4** | 0 | 0 | 0 | 0 |
| **Neutrophils (x10$^9$/L)** | **1.18** | 2.08 | **1.79** | 5.78 | 2.27 | 7.78 | 5.88 | 3.52 | **1.3** | 3.94 | 1.85 |
| **Eosinophils (x10$^9$/L)** | **0** | **0** | 0.03 | 0.42 | 0.03 | **0** | 0.08 | 0.05 | 0.02 | 0.05 | **0** |
| **Basophils (x10$^9$/L)** | **0** | **0** | 0.03 | 0.08 | 0.03 | **0** | **0** | **0** | **0** | **0** | **0** |
| **Monocytes (x10$^9$/L)** | 0.11 | 0.35 | 0.16 | 0.67 | 0.14 | 0.42 | 0.4 | 0.33 | 0.09 | 0.38 | 0.21 |
| **Lymphocytes (x10$^9$/L)** | **0.82** | 2.64 | 1.18 | 1.42 | **0.27** | 2.31 | 1.37 | 1.52 | **0.47** | **1.09** | **0.92** |
| **Bands (x10$^9$/L)** | 0 | 0 | 0 | 0 | 0 | 0 | 0.32 | 0 | 0 | 0 | 0 |
| **AST (U/L)** | **66** | 28 | **184** | 81 | **105** | 17 | 27 | 21 | **39** | 44 | **69** |
| **ALT (U/L)** | 26 | **9** | 314 | 47 | **86** | **15** | 24 | 17 | 22 | 30 | 62 |
| **Clinical WHO Classification** | DWS | DWW | SD | DWW | SD | DWS | DWW | DWS | DWS | DWS | DWW |

The study included a total of 11 dengue patients (5 females and 6 males), classified as dengue without warning signs (DWW, n = 4), dengue with warning signs (DWS, n = 5), or severe dengue (SD, n = 2), according to WHO criteria.

Abbreviations: %, percentage; fL, femtoliter; g/L, grams/Liter; L, liter; pg, picograms; RBC, red blood cells; WBC, white blood cells. Hgb hemoglobin; Hct hematocrit; RBC red blood cell count; RDW red cell distribution width; MCV mean cell volume; MCH mean corpuscular hemoglobin; MCHC mean corpuscular hemoglobin concentration; WBC white blood cell count; PLT platelet count; MPV mean platelet volume; AST: Aspartate transaminase; ALT: Alanine transaminase; DWW: dengue without warning signs; DWS: dengue with warning signs and SD: severe dengue.

**Table 2. Reference virus strains.**

| Virus isolate | GenBank accession number | Abbreviation |
|---|---|---|
| DENV1 | AY145121.1 | DENV1 |
| DENV4 | AY648301 | DENV4 |
| ZIKV | KX377337.1 | ZIKV PR |

**Table 3. Flavivirus clinical isolates.**

| Virus isolate | GenBank accession number | Abbreviation |
|---|---|---|
| DENV1 | AY145121.1 | DENV1 |
| DENV4 | AY648301 | DENV4 |
| ZIKV | KX377337.1 | ZIKV PR |

NA- Not applicable.

Supernatants were harvested at peak titer, clarified by centrifugation at 2000 rpm for 10 min, and stabilized with SPG buffer (2.18 mM sucrose, 38 mM monobasic $K_2HPO_4$, 72 mM dibasic $K_2HPO_4$, 60 mM L-glutamic acid; all from Sigma-Aldrich, St. Louis, MO, USA). Aliquots were homogenized and cryopreserved at −80 °C until further use.

## Viral titration by focus-forming assay

Viral infectivity was determined using an immunodetection-based focus-forming assay (FFA). Briefly, Vero cells were infected with serial dilutions of viral stocks, and infected foci were detected using the pan-flavivirus monoclonal antibody 4G2. This assay was used to quantify viral stocks and to normalize infectious input prior to infection experiments.

Ten-fold serial dilutions of viral stocks or infected cell supernatants were inoculated onto confluent Vero cells in 24-well plates. After 1 h adsorption, inocula were removed and monolayers washed with PBS. Cells were then overlaid with RPMI containing 1% methylcellulose (Sigma-Aldrich), 2% fetal bovine serum (FBS; Biowest), and 1% antibiotic/antimycotic solution (Biowest). After 5 days at 37 °C and 5% $CO_2$, monolayers were washed and fixed/permeabilized with 80% methanol for 30 min. Fixed cells were blocked with 5% low-fat milk in PBS for 15 min and immunostained with mouse anti-pan-flavivirus monoclonal antibody 4G2 (kindly provided by Paola Carolina Valenzuela León, NIH, MD, USA). After rinsing, plates were incubated with HRP-conjugated rabbit anti-mouse IgG (H+L) (KPL). Foci were developed using TrueBlue peroxidase substrate (KPL) and counted manually under a stereoscopic microscope. Titers were expressed as $\log_{10}$ FFU/mL.

## Flow cytometry

Precursor-derived PLP-associated dengue E protein was detected by immunostaining. Infected cells, platelet-like particles (PLPs), and patient platelets were incubated with fixation and permeabilization buffer according to the manufacturer's instructions (eBioscience). Cells were then incubated with monoclonal antibody 4G2 diluted in PBS supplemented with 1% fetal bovine serum (PBS/FBS) for 1 h at room temperature. After two washes with PBS/FBS, cells were incubated with Alexa Fluor 488–conjugated rabbit anti-mouse secondary antibody (Thermo Fisher Scientific). Following final washes in PBS/FBS, samples were analyzed on an Attune NxT flow cytometer (Thermo Fisher Scientific).

For surface marker detection, megakaryocytic precursor cell lines and PLPs were stained for CD41. Cells were first incubated with Fc-blocking solution (Human TruStain, BioLegend), followed by Alexa Fluor 647–conjugated mouse anti-human CD41 antibody (BioLegend) for 30 min. Cells were washed with PBS/FBS and analyzed on the Attune NxT cytometer. Flow cytometry data were analyzed using FlowJo version 10.9.

PLPs were identified based on their forward and side scatter properties using size calibration beads and by expression of the platelet marker CD41. Events outside the defined platelet/PLP gate and CD41-negative events were excluded to minimize inclusion of cellular debris or apoptotic fragments.

## Differentiation of megakaryocytic precursors and PLP production

Differentiation of MEG-01 and K562 cells was performed based on the protocol of Banerjee et al., with modifications. Briefly, $2 \times 10^5$ cells were seeded and stimulated with 25 ng/mL PMA for 3, 4, and 7 days. Cells were harvested and analyzed for the megakaryocyte-specific marker CD41 using flow cytometry [13]. Supernatants from K562 cultures were clarified by two sequential centrifugations (150 g for 15 min and 750 g for 15 min). PLPs were pelleted at 1600 g for 15 min, resuspended, and immunostained for viral E protein or surface CD41 as described above.

For differentiation assays, phorbol 12-myristate 13-acetate (PMA) was added simultaneously with viral infection and maintained throughout the culture period. PMA treatment was used as a differentiation control and was not applied as a pre-conditioning stimulus prior to infection.

## Isolation and flow cytometry analysis of platelets from dengue patients

Peripheral blood samples were collected from laboratory-confirmed dengue patients during the 2022 outbreak in Oaxaca city. Diagnosis was confirmed by dengue antigen immunochromatography, and disease severity was classified according to WHO guidelines using laboratory data (platelet count, hematocrit, liver enzyme levels) and clinical criteria [3]. Patient demographics and laboratory findings are summarized in Table 1.

Clinical metadata regarding the exact number of days between symptom onset and sample collection, as well as quantitative viremia at the time of isolation, were not consistently available for all patients and therefore could not be included in the analysis.

Platelet-rich plasma (PRP) was separated by centrifugation at 150 g for 15 min, mixed with an equal volume of CGS-EDTA buffer (1.29 mM sodium citrate, 3.12 mM citric acid, 4.99 mM EDTA, pH 6.5), and centrifuged for 20 min at 100 g. Platelets were recovered by centrifugation of the supernatant at 500 g for 15 min. The platelet pellet was resuspended in 3 mL CGS-EDTA buffer, centrifuged at 500 g for 5 min, and stained for E protein or CD41 following the same immunostaining protocol described above.

This isolation protocol is based on previously established procedures designed to minimize platelet activation during flow cytometric analysis [16].

## Statistical analysis

Differences in percentages and mean fluorescence intensity (MFI) from flow cytometry data were analyzed by one-way ANOVA followed by Tukey-multiple comparisons test, unless otherwise indicated. For comparisons between two independent groups, an unpaired two-tailed Student's t-test was used.

Replication kinetics were analyzed using two-way repeated-measures ANOVA, with virus and time as factors, followed by Bonferroni's post hoc multiple-comparison test to account for repeated measurements over time.

Correlation analyses were performed using Spearman's rank correlation coefficient (two-tailed), given the limited number of independent infection conditions and to avoid assumptions of normal data distribution. For correlation analyses (Figs 5 and 10), each data point represents one independent infection condition (one viral isolate/experimental condition), plotted as the mean of biological replicates.

For analyses involving platelet data from human donors, individual data points were visualized to reflect inter-individual variability. Due to limited sample size and non-normal distribution of clinical data, comparisons between healthy donors and dengue patient groups were performed using the non-parametric Mann–Whitney U test. Group medians are shown in the Figs.

All experiments were performed using at least three independent biological replicates unless otherwise indicated. Biological replicates were defined as independent cell cultures or independent patient samples, as appropriate. Statistical analyses were conducted using GraphPad Prism (version 10). Data are presented as mean ± standard deviation (SD), unless otherwise specified. When applicable, statistically distinct groups were indicated using letter-based annotations following post hoc testing.

## Results

### Permissivity of megakaryocytic precursors to DENV infection

To characterize the permissiveness and replication kinetics of flaviviruses in megakaryocytic precursor models, MEG-01 and K562 cells were infected with reference strains of DENV and ZIKV and viral infection was evaluated by intracellular E protein detection and infectious virus production.

At 3 days post-infection (dpi), viral E protein was detected by flow cytometry in both cell lines infected with DENV1, whereas ZIKV PR infection resulted in lower levels of E+ cells (Fig 1A). In MEG-01 cells infected at a multiplicity of

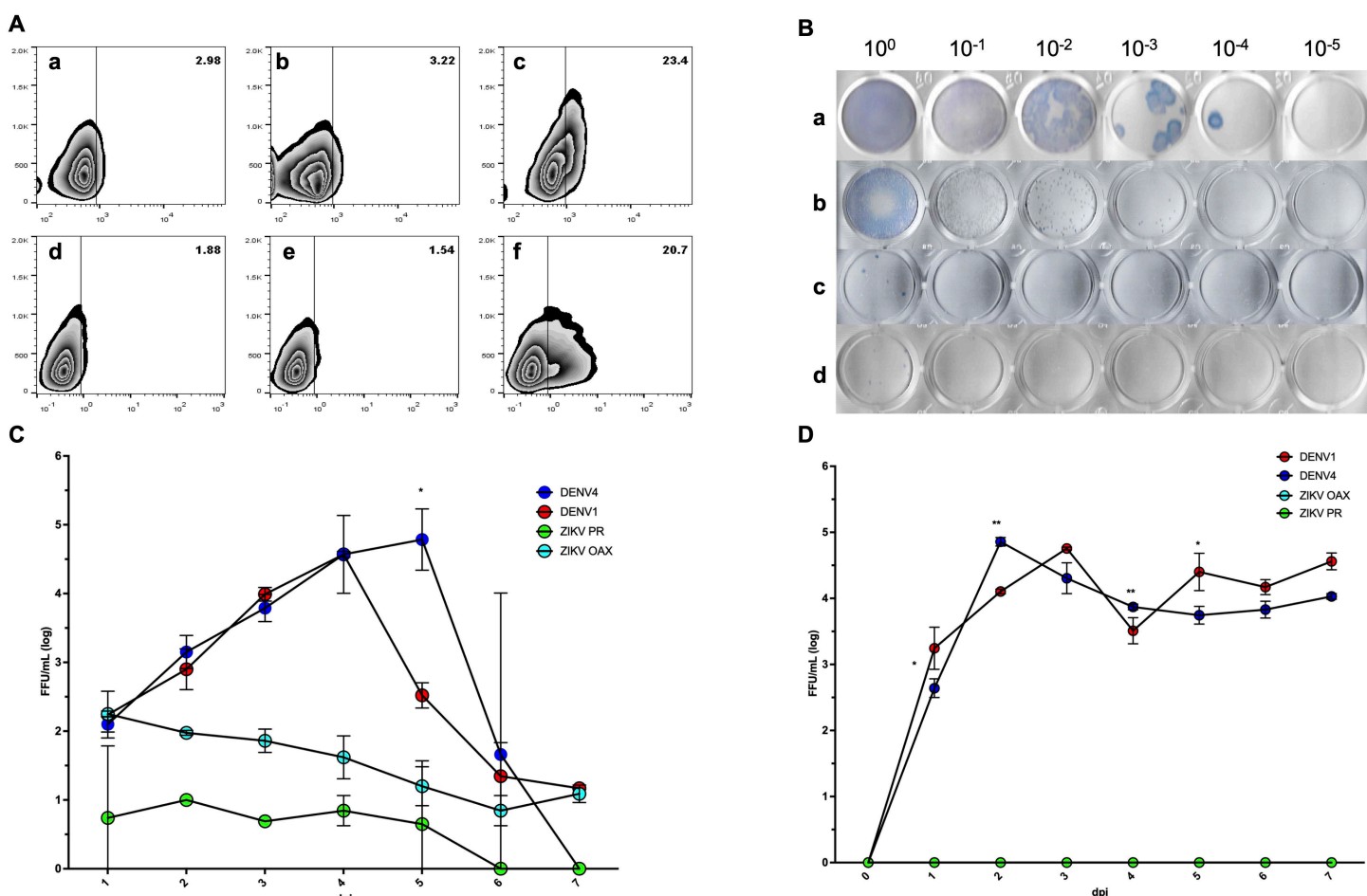

**Fig 1. Permissiveness and replication kinetics of flaviviruses in megakaryocytic precursor cell lines.** Intracellular viral envelope (E) protein expression and infectious virus production were evaluated in MEG-01 and K562 cells following infection with reference strains of dengue virus (DENV) and Zika virus (ZIKV). **(A)** Representative flow cytometry contour plots showing intracellular E protein detection at 3 days post-infection (dpi). MEG-01 cells were infected at a multiplicity of infection (MOI) of 0.1 and are shown as mock-infected controls (panel **a**), ZIKV PR–infected cells (panel **b**), and DENV1-infected cells (panel **c**). K562 cells were infected at MOI 1 and are shown as mock-infected controls (panel **d**), ZIKV PR–infected cells (panel **e**), and DENV1-infected cells (panel **f**). **(B)** Focus-forming assays of clarified supernatants collected at 3 dpi, showing infectious virus production from MEG-01 cells infected with DENV1 (panel **a**), K562 cells infected with DENV1 (panel **b**), MEG-01 cells infected with ZIKV PR (panel **c**), and K562 cells infected with ZIKV PR (panel **d**), using the same MOIs described in panel A (MEG-01, MOI 0.1; K562, MOI 1). **(C)** Seven-day replication kinetics of DENV1, DENV4, ZIKV PR, and ZIKV OAX in MEG-01 cells infected at MOI 0.1. **(D)** Seven-day replication kinetics of the same viral strains in K562 cells infected at MOI 1. Viral titers are expressed as focus-forming units per milliliter (FFU/mL). Data represent mean±SD from independent experiments. Statistical significance across time points was assessed by two-way repeated-measures ANOVA with Bonferroni correction.

infection (MOI) of 0.1, DENV1 induced a clear E⁺ population (Fig 1A panel c), compared with mock-infected controls (Fig 1A panel a), while ZIKV PR produced a weaker signal (Fig 1A, panel b). Similarly, K562 cells infected at MOI 1 showed detectable E protein following DENV1 infection (Fig 1A panel f), compared with mock-infected controls (Fig 1A panel d), whereas ZIKV PR infection resulted in limited E expression (Fig 1A, panel e).

Infectious virus production was confirmed by focus-forming assays of clarified culture supernatants collected at 3 dpi. Both MEG-01 (Fig 1B, panel a) and K562 cells (Fig 1B, panel b) produced infectious DENV1 particles. In contrast, ZIKV PR replication was minimal or undetectable under the same conditions (Fig 1B, panels c and d).

To further assess viral replication dynamics, full 7-day replication kinetics were performed. In MEG-01 cells infected at MOI 0.1, DENV1 and DENV4 exhibited productive replication, reaching peak titers between 4 and 5 dpi, followed by a decline at later time points (Fig 1C). ZIKV PR and ZIKV OAX showed substantially lower replication levels throughout the time course. In K562 cells infected at MOI 1, both DENV1 and DENV4 replicated efficiently, reaching higher peak titers between 2 and 3 dpi, while ZIKV strains remained poorly replicative (Fig 1D).

Together, these data demonstrate that both MEG-01 and K562 cells are permissive to DENV infection, with K562 cells supporting higher levels of viral replication. These findings validate the use of these cell lines as in vitro models to study downstream effects of DENV infection on megakaryocytic differentiation and platelet-like particle generation.

## Effect of DENV infection on differentiation markers in MEG-01 cells

The effect of DENV1 infection on MEG-01 differentiation was analyzed by surface CD41 expression. At 4 and 7 dpi, both the percentage of CD41⁺ cells and CD41 MFI were significantly higher in DENV1-infected cultures compared with basal conditions (Fig 2A–B; one-way ANOVA with Tukey's post-hoc test, $p < 0.05$). At 7 dpi, CD41 MFI was significantly higher in DENV1-infected cells compared with PMA-treated cultures, whereas no significant difference was observed relative to basal conditions. PMA stimulation served as a positive control and increased CD41 expression as expected. Notably, combined PMA and DENV1 treatment did not result in additive increases in CD41 expression, suggesting that pharmacological differentiation and viral infection may differentially modulate megakaryocytic marker expression over time.

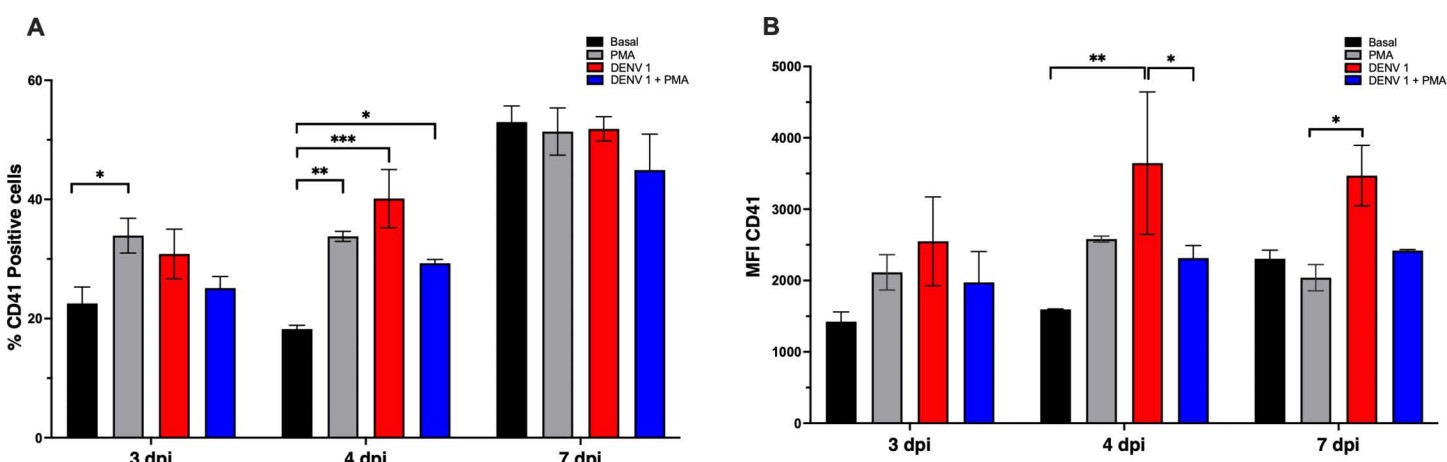

**Fig 2. Effect of DENV1 infection on MEG-01 differentiation. (A)** Percentage of CD41⁺ MEG-01 cells at 3, 4, and 7 days post-infection (dpi) under basal conditions, pharmacological stimulation with PMA, infection with DENV1, or PMA plus DENV1. **(B)** CD41 MFI under the same conditions and time points. Data are presented as mean ± SD of three independent biological replicates (n = 3). Statistical significance was evaluated by one-way ANOVA followed by Tukey's multiple-comparisons test (p < 0.05). Different letters above bars indicate statistically distinct groups; groups sharing a letter are not significantly different.

## Effect of DENV infection on differentiation markers in K562 cells

Based on initial analyses showing comparable qualitative effects of DENV infection on CD41 expression in MEG-01 cells (Fig 2), subsequent experiments evaluating isolate-dependent effects and infection kinetics were performed using K562 cells. K562 cells represent an earlier megakaryocyte–erythroid progenitor stage and exhibited greater experimental robustness across multiple viral isolates and time points, enabling consistent comparative analyses.

K562 cells displayed isolate-dependent effects on differentiation. At 3 dpi, the Oax-2019–2 and Oax-2019–4 isolates induced significantly higher percentages of CD41$^+$cells (Fig 3A) and increased CD41 MFI (Fig 3D) compared with basal conditions (p < 0.05). At 4 dpi, the isolate Oax-2019–4 maintained significantly higher percentages of CD41$^+$cells (Fig 3B) and CD41 MFI (Fig 3E). By 7 dpi, all DENV isolates tested induced significantly higher CD41$^+$percentages and/or CD41 MFI compared with basal conditions (Fig 3C and F), whereas ZIKV PR and ZIKV Oax did not induce significant changes. Statistical differences between groups are indicated by distinct letters above the bars, as determined by post-hoc comparisons (p < 0.05).

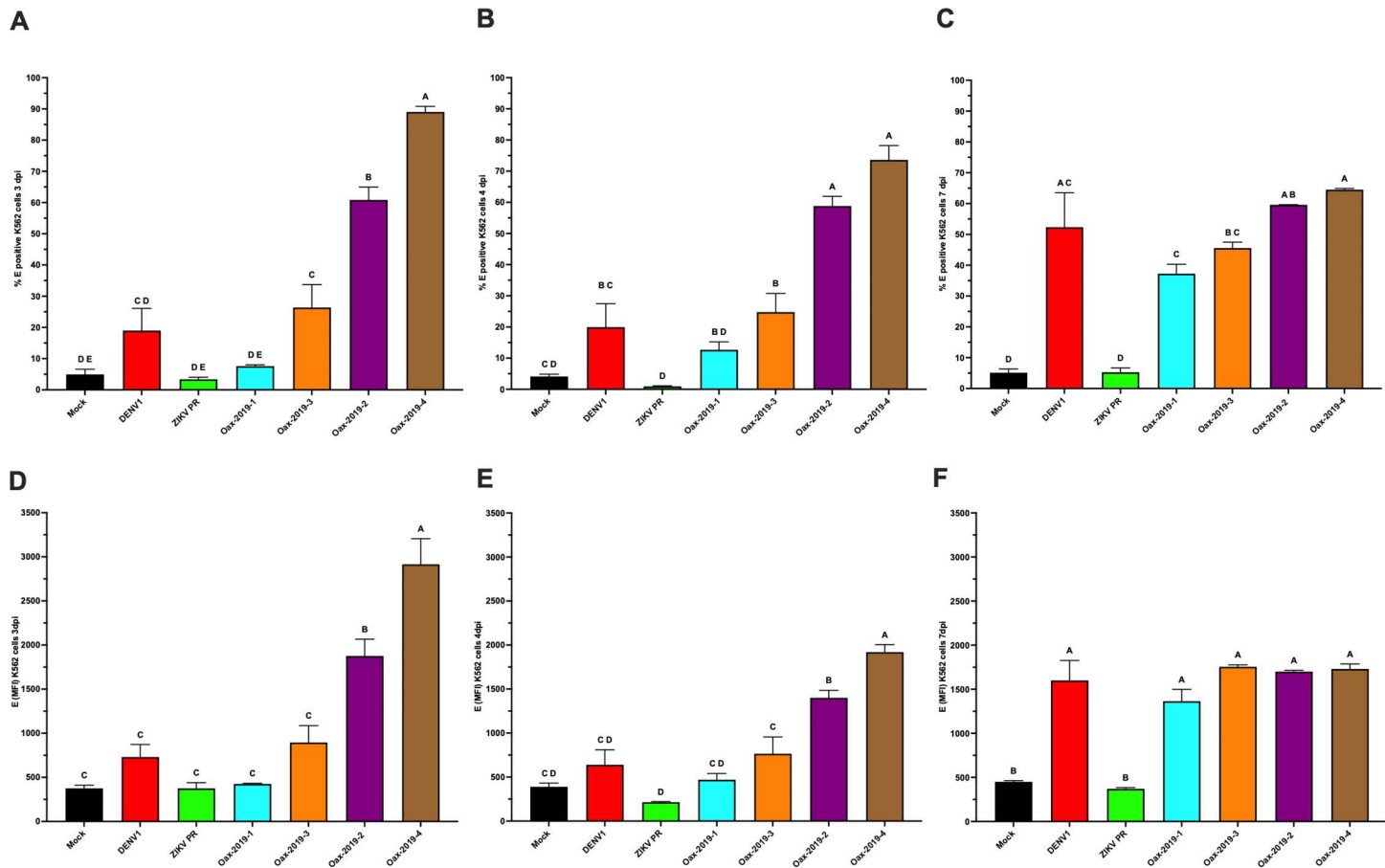

**Fig 3.  Differentiation of K562 cells after infection with DENV or ZIKV isolates. (A–C)** Percentage of CD41$^+$K562 cells at 3, 4, and 7 dpi under basal conditions, PMA stimulation, or infection with DENV1, ZIKV PR, or DENV2 clinical isolates (Oax-2019-1, Oax-2019-2, Oax-2019-3, Oax-2019-4). **(D–F)** CD41 MFI at the same time points under the same conditions. Data are presented as mean ± SD of three independent biological replicates (n = 3). Statistical significance was evaluated by one-way ANOVA followed by Tukey's multiple-comparisons test (p < 0.05). Different letters above bars indicate statistically distinct groups; groups sharing a letter are not significantly different.

## Infection kinetics of K562 cells with multiple DENV isolates

To evaluate whether infection of megakaryocyte–erythroid precursors is isolate dependent, K562 cells were infected at MOI 1 with a DENV1 reference strain and with four DENV2 isolates derived from patients with different thrombocytopenia levels. At 3 dpi, the Oax-2019–3, Oax-2019–2, and Oax-2019–4 isolates induced a significant increase in the percentage of E$^+$K562 cells compared with mock controls (Fig 4A). At 4 dpi, these isolates, together with DENV1 reference strain, continued to show significantly higher percentages of E$^+$ cells (Fig 4B). By 7 dpi, all DENV isolates tested showed significantly increased percentages of E$^+$ cells compared with controls (Fig 4C).

Analysis of precursor-derived PLP-associated E protein levels showed a similar pattern. At 3 and 4 dpi, the Oax-2019–2 and Oax-2019–4 isolates induced significantly higher E MFI compared with mock and ZIKV-infected cultures (Figs 4D–E). At 7 dpi, all DENV isolates produced significantly elevated E MFI values compared with controls (Fig 4F). In contrast to the experiments shown in Fig 1, which directly assessed viral replication kinetics, Fig 4 focuses on isolate-dependent differences in infection outcomes following standardized infectious input. Together, these results confirm that K562 cells are permissive to multiple DENV isolates, while revealing isolate-dependent variation in infection-associated E protein detection.

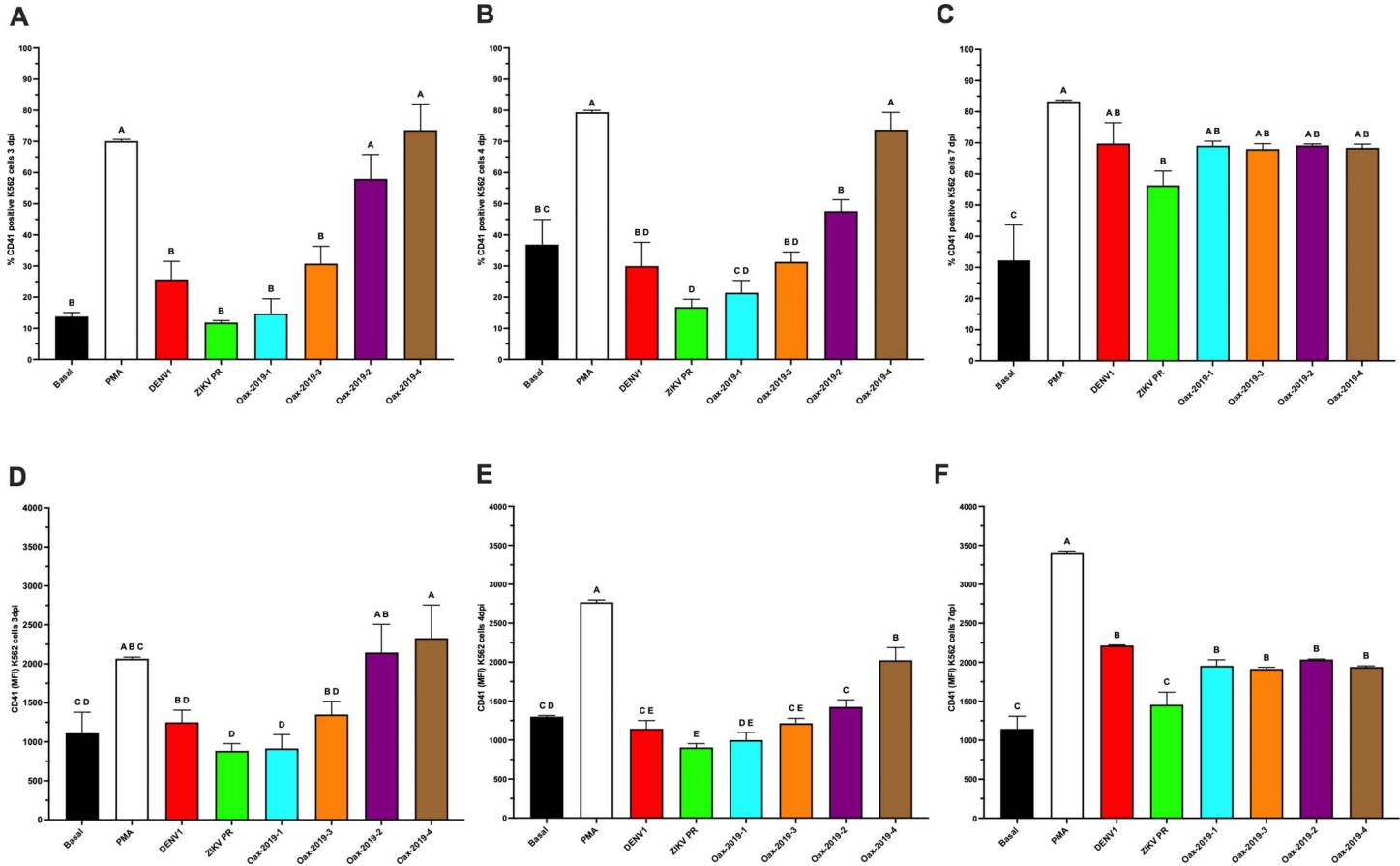

**Fig 4. Infection kinetics of K562 cells with DENV and ZIKV isolates. (A–C)** Percentage of E$^+$K562 cells at 3, 4, and 7 dpi. **(D–F)** E protein MFI in infected K562 cells at the same time points. Data are presented as mean±SD of three independent biological replicates (n=3). Statistical significance was evaluated by one-way ANOVA followed by Tukey's multiple-comparisons test (p<0.05). Different letters above bars indicate statistically distinct groups; groups sharing a letter are not significantly different.

## Kinetics of infection and association with early differentiation

The relationship between infection and differentiation in K562 cells was evaluated by correlating the percentage of E⁺ cells with the percentage of CD41⁺ cells at matched time points. At 3 dpi, a strong positive correlation was observed (r = 0.9972, P < 0.0001; Fig 5A). A significant positive correlation was detected at 4 dpi (r = 0.7857, P = 0.0480; Fig 5B). At 7 dpi, the correlation was not significant (r = 0.6786, P = 0.1095; Fig 5C), suggesting that the infection-associated differentiation effect is more evident at early stages and attenuates over time.

## Production of platelet-like particles and detection of E protein in vitro

Supernatants from DENV-infected K562 cultures were clarified and analyzed for PLP production. At 3 dpi, the percentage of CD41⁺ PLPs and CD41 MFI were significantly increased in several isolates compared with basal conditions (Fig 6A, 6D; p < 0.05). At 4 dpi, PLP percentages converged among conditions, while CD41 MFI remained elevated in infected cultures (Fig 6B, 6E). At 7 dpi, most groups reached comparable PLP percentages, although CD41 MFI remained higher in some DENV-infected cultures (Fig 6C, 6F).

To evaluate whether megakaryocytic infection leads to the generation of PLPs carrying viral antigen, the presence of E protein in CD41⁺ PLPs was analyzed in supernatants from DENV isolate–infected K562 cultures. At 3 dpi, several DENV isolates induced a significant increase in the percentage of CD41⁺E⁺ PLPs and in E protein MFI compared with mock controls (Fig 7A and 7D; one-way ANOVA with Tukey's post hoc test, p < 0.05). At 4 dpi, differences among conditions were less pronounced and did not reach statistical significance (Fig 7B and 7E). At 7 dpi, although some isolates displayed higher mean values, increased variability prevented consistent statistically significant differences in either the percentage of CD41⁺E⁺ PLPs or E MFI (Fig 7C and 7F).

## Ex vivo exposure of donor platelets to DENV1

To assess whether mature platelets can acquire dengue E protein by direct exposure, peripheral blood platelets from a healthy donor were incubated with DENV1 at multiplicities of infection (MOI) 1 or 3 for 1 or 2 days. Flow cytometry analysis following fixation and permeabilization revealed no significant differences in either the percentage of E⁺ platelets or mean fluorescence intensity (MFI) of E protein compared with mock controls (Figs 8A–B; n = 3)

To assess whether mature platelets can acquire dengue virus E protein through direct exposure, peripheral blood platelets from a healthy donor were incubated with DENV1 at multiplicities of infection (MOI) of 1 or 2

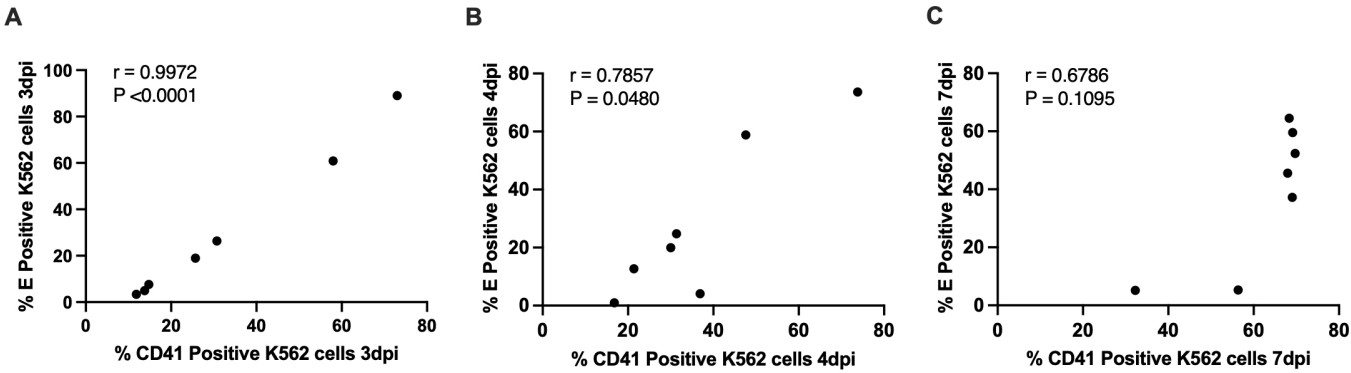

**Fig 5. Correlation between infection and differentiation of K562 cells.** Spearman's correlation between the percentage of E⁺ K562 cells and the percentage of CD41⁺ K562 cells at (A) 3 dpi, (B) 4 dpi, and (C) 7 dpi. Each data point represents one independent infection condition (one viral isolate), plotted as the mean of biological replicates. Spearman's r and two-tailed p values are in each panel.

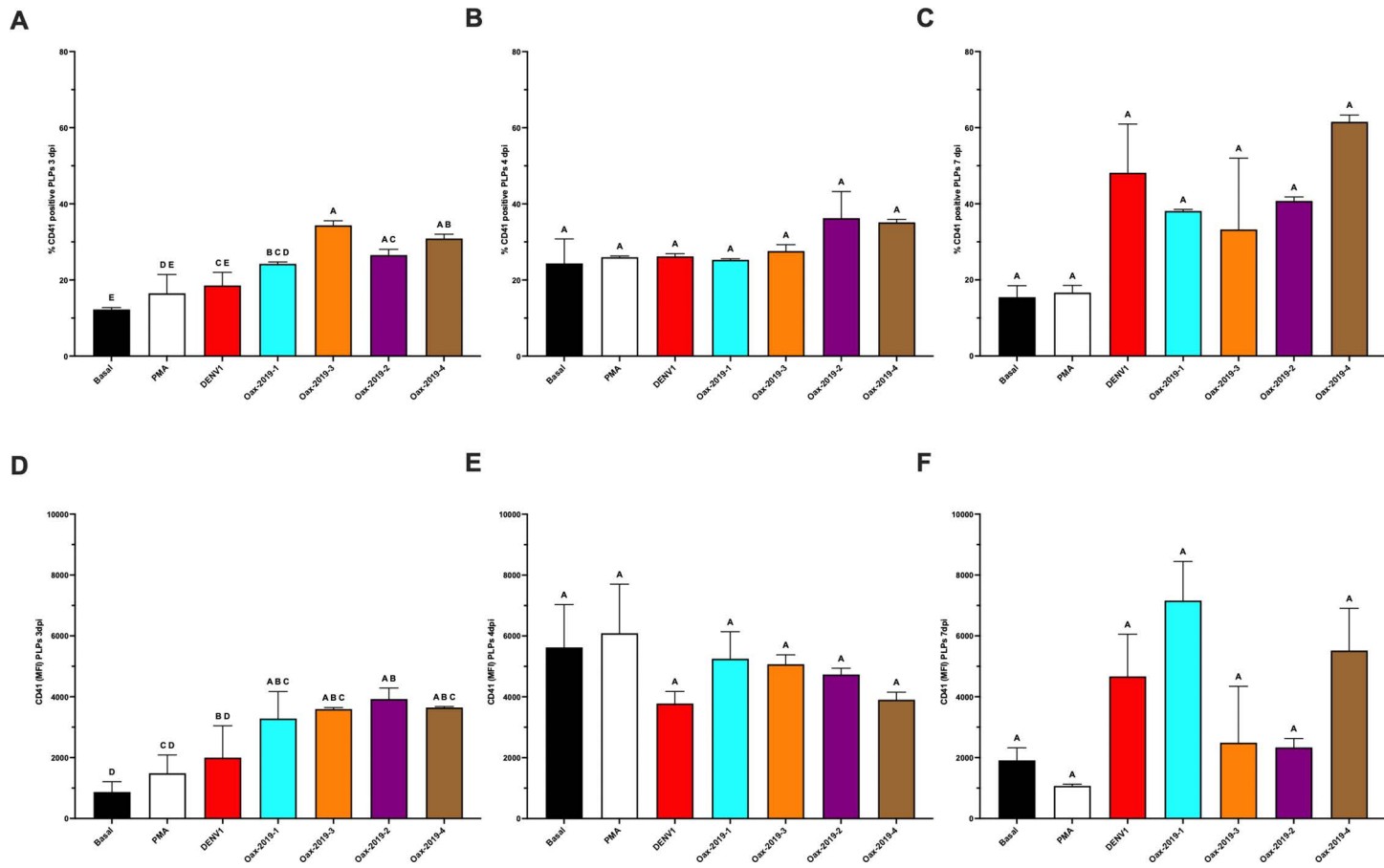

**Fig 6. Production of platelet-like particles (PLPs) in K562 cell supernatants after DENV infection or PMA stimulation. (A–C)** Percentage of CD41$^+$PLPs at 3, 4, and 7 days post-infection (dpi). **(D–F)** CD41 mean fluorescence intensity (MFI) in PLPs at the same time points. PLP values are expressed as percentages of total events to normalize variability in culture yield and cytometer acquisition; absolute PLP counts were not consistently available across experiments. Data are presented as mean±SD of three independent biological replicates (n=3). Statistical significance was evaluated by one-way ANOVA followed by Tukey's multiple-comparisons test (p<0.05). Different letters above bars indicate statistically distinct groups; groups sharing a letter are not significantly different.

days. Flow cytometry analysis following fixation and permeabilization revealed no significant differences in either the percentage of E$^+$ platelets or the mean fluorescence intensity (MFI) of E protein compared with mock-treated controls (Figs 8A–B; n=3).

### Detection of E protein in platelets from dengue patients

Platelets from dengue patients showed significantly higher percentages of E$^+$ cells and increased E protein mean fluorescence intensity (MFI) compared with platelets from healthy donors (Fig 9). When patients were grouped according to clinical severity, no statistically significant differences were detected between dengue without warning signs (DWW) and dengue with warning signs or severe dengue (DWS/SD).

Nevertheless, a trend toward higher proportions of E$^+$ platelets was observed in patients with greater clinical severity, particularly within the DWS/SD group.

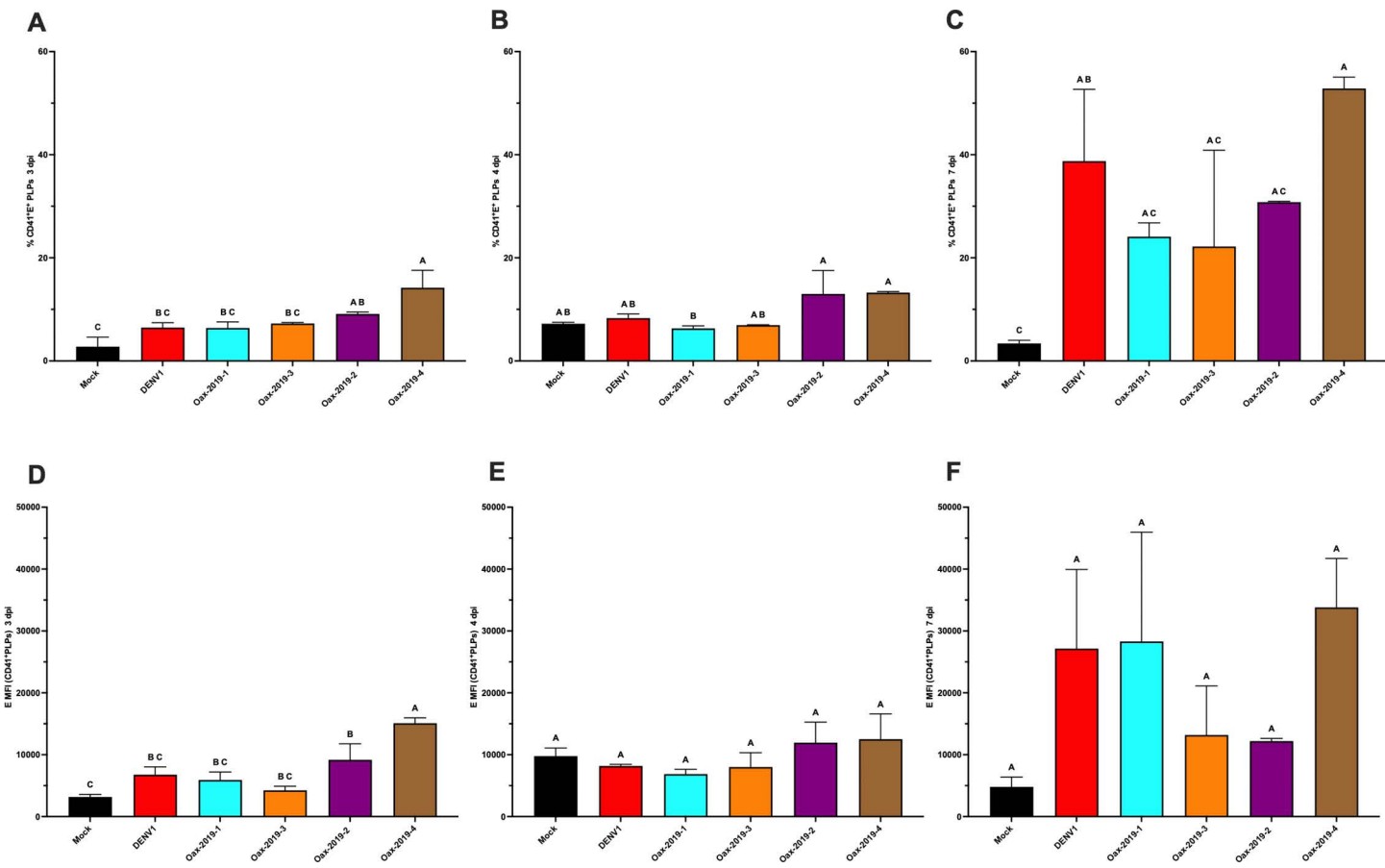

**Fig 7. Detection of dengue E protein in PLPs from infected K562 cells. (A–C)** Percentage of CD41⁺E⁺PLPs at 3, 4, and 7 dpi. **(D–F)** E protein MFI in CD41⁺PLPs at the same time points. Data are presented as mean ± SD of three independent biological replicates (n = 3). Statistical significance was evaluated by one-way ANOVA followed by Tukey's multiple-comparisons test (p < 0.05). Different letters above bars indicate statistically distinct groups; groups sharing a letter are not significantly different.

## Correlation between megakaryocytic precursor infection and the generation of E protein–positive platelet-like particles in vitro

To assess the relationship between megakaryocytic precursor infection and the generation of E protein–positive PLPs in vitro, correlation analyses were performed using Spearman's rank test. At 3 dpi, a significant positive correlation was observed between the percentage of E⁺ K562 cells and the percentage of CD41⁺E⁺PLPs generated from the same cultures (Fig 10A; r = 0.7857, P = 0.048). At 4 dpi, a positive trend was observed that did not reach statistical significance (Fig 10B; r = 0.75, P = 0.0663). At 7 dpi, a positive trend between precursor infection and PLP-associated E protein levels was observed; however, the correlation did not reach statistical significance (Fig 10C). Increased variability at later time points may contribute to this result.

## Discussion

In this study, we demonstrate that DENV can infect megakaryocytic precursors and that this infection is associated with accelerated differentiation and the transfer of viral E protein to the PLPs they release. While K562 cells are widely used as a model to study antibody-dependent enhancement of dengue virus infection, their permissivity to direct infection

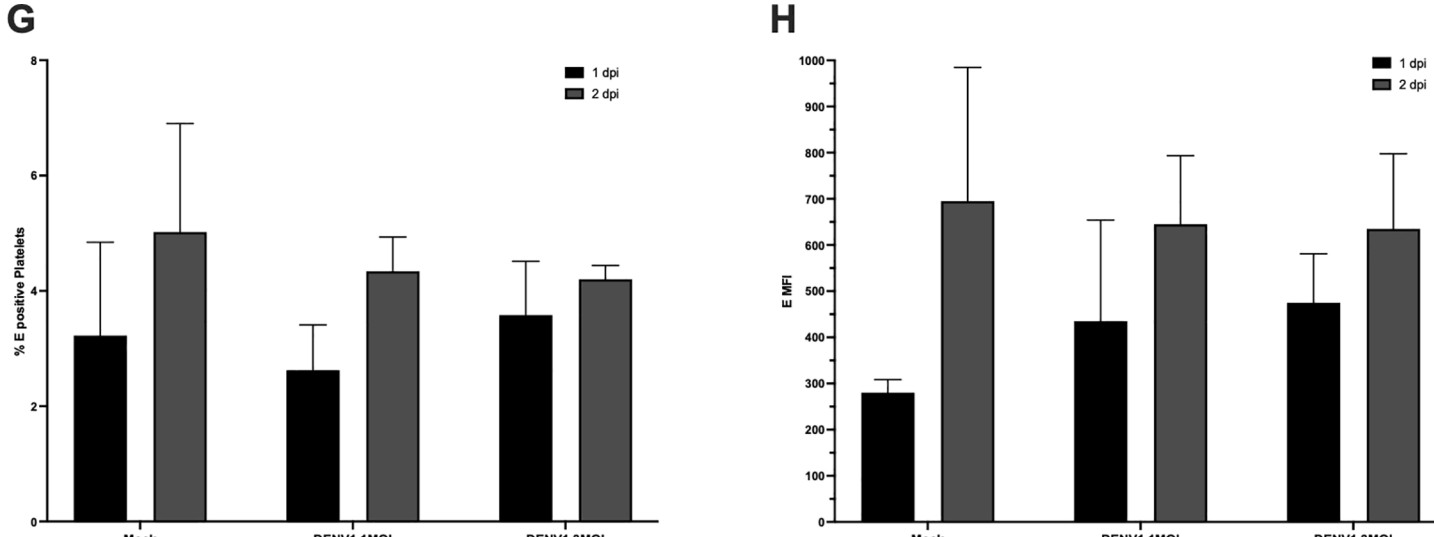

**Fig 8. Lack of E protein uptake by platelets incubated ex vivo with DENV1.** Peripheral blood platelets from a healthy donor were incubated with DENV1 at MOI 1 or MOI 3 for 1 or 2 days. **(A)** Percentage of E+ platelets. **(B)** E protein MFI. Data are presented as mean ± SD of three independent biological replicates (n = 3). Statistical significance was evaluated by one-way ANOVA followed by Tukey's multiple-comparisons test (p < 0.05). No significant differences were detected between DENV1-exposed conditions and mock controls.

has also been reported [11]. In this study, K562 cells were employed to model megakaryocytic precursor infection under antibody-independent conditions, allowing assessment of infection-associated differentiation and antigen transfer without the confounding effects of immune complexes. Across multiple readouts, including PLP production and E protein incorporation, DENV isolate – dependent effects were most evident at early time points, with later stages showing convergence across conditions. Correlation analyses using Spearman's rank coefficient revealed significant associations between precursor infection and differentiation markers at early time points, as well as between precursor infection and PLP-associated E protein levels. At later stages, although positive trends were observed, some correlations did not reach statistical significance, likely reflecting increased biological variability among isolates over time. These findings suggest that DENV infection primarily influences early megakaryocytic differentiation and antigen transfer rather than inducing sustained alterations in platelet-like particle composition. Importantly, we show that platelet-associated E protein can be detected in platelets from dengue patients. Although differences between dengue clinical subgroups did not reach statistical significance, platelets from dengue patients consistently exhibited higher levels of dengue virus E protein compared with healthy donors. The absence of significant differences between DWW and DWS/SD groups likely reflects the limited sample size and inter-individual variability inherent to clinical cohorts.

Importantly, the consistent detection of E+ platelets in infected individuals, but not in healthy donors, supports the presence of platelet-associated viral antigens during dengue infection. The observed trend toward higher proportions of E+ platelets in patients with greater clinical severity should therefore be interpreted with caution and warrants further investigation in larger, prospectively characterized cohorts.

Our results extend prior observations by Nakao et al., who identified DENV RNA in immature bone marrow progenitors [9] and Clark et al., who showed that DENV2 can infect megakaryocytic-erythroid progenitors (MEPs) [10]. These studies suggested that precursors may act as sites of viral replication. We add to this by showing that infection coincides with changes in differentiation (CD41 expression) and the generation of PLPs carrying viral proteins, and by linking these events to patient samples.

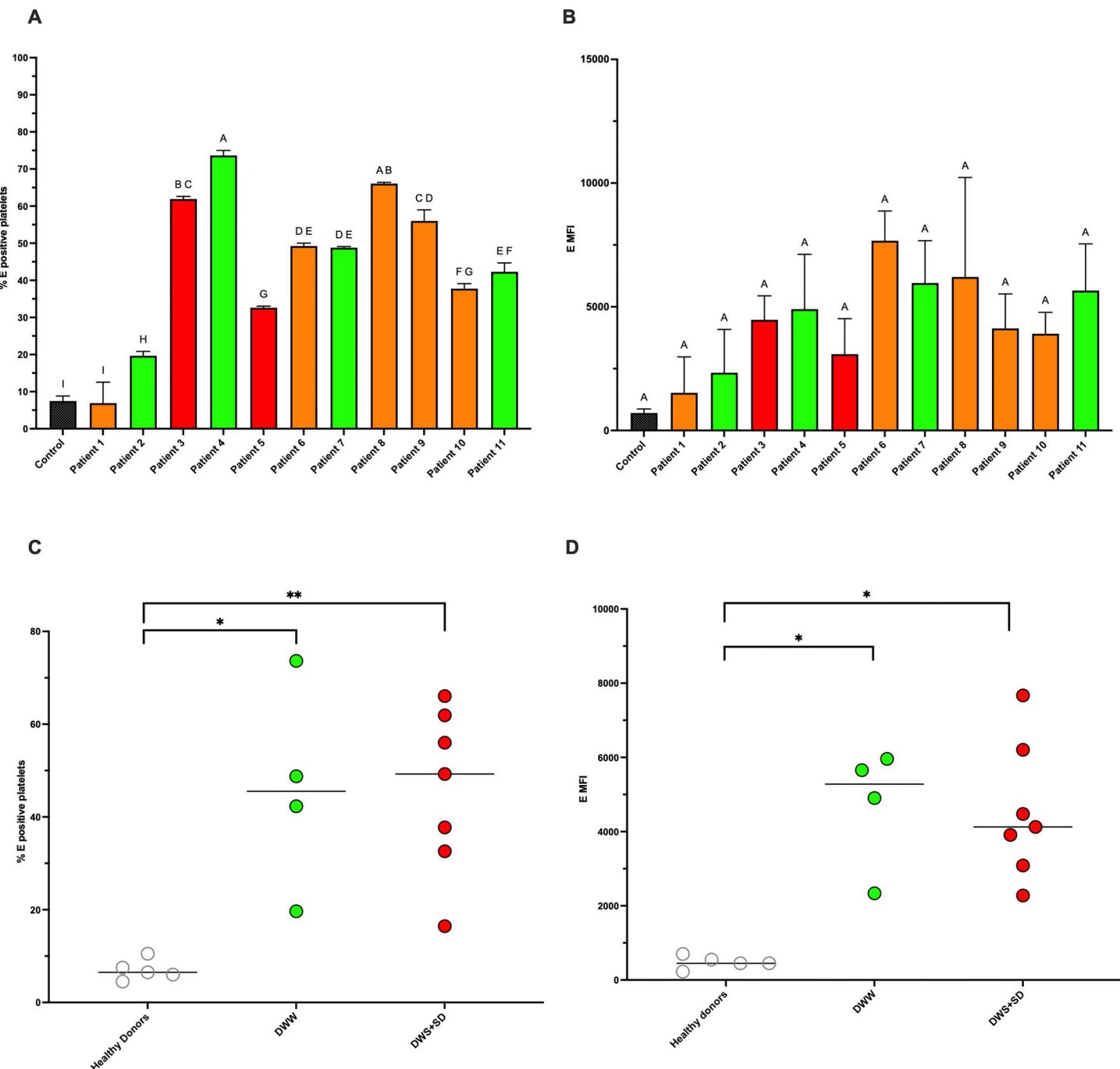

**Fig 9. Detection of dengue virus E protein in platelets from dengue patients and healthy donors.** Peripheral blood platelets were analyzed by flow cytometry for dengue virus envelope (E) protein expression. **(A)** Percentage of E+ platelets and **(B)** E protein mean fluorescence intensity (MFI) in healthy donors, patients with dengue without warning signs (DWW), and patients with dengue with warning signs or severe dengue (DWS/SD). Each dot represents an individual donor or patient sample; horizontal lines indicate group medians. Statistical comparisons were performed using the non-parametric Mann–Whitney U test. Significant differences were detected between dengue patient groups and healthy donors (p < 0.05; p < 0.01). No statistically significant differences were observed between DWW and DWS/SD groups.

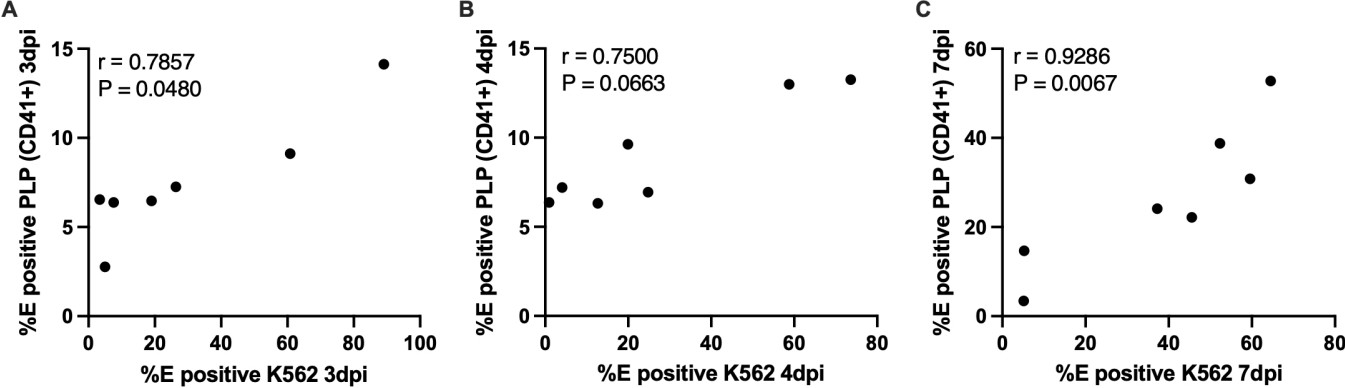

**Fig 10. Correlation between megakaryocytic precursor infection and the generation of E protein–positive platelet-like particles in vitro.** Spearman correlation between the percentage of E$^+$ K562 cells and the percentage of CD41$^+$E$^+$PLPs generated from the same cultures at (A) 3 dpi, (B) 4 dpi, and (C) 7 dpi. Each data point represents an independent in vitro infection condition (one viral isolate), plotted as the mean of biological replicates. Spearman's *r* and two-tailed *p* values are shown in each panel.

The absence of E$^+$ platelets following ex vivo exposure of primary platelets to DENV suggests that platelet-associated E protein in patients is unlikely to arise from passive surface adsorption or direct infection of circulating platelets. Instead, these findings support a precursor-based origin of platelet-associated viral antigen during megakaryopoiesis. Importantly, the biological implications of platelet-associated E protein are likely independent of whether the antigen is retained in intra-cellular compartments or associated with membrane structures following transfer from infected precursors. The reduced CD41 MFI observed in the PMA+DENV condition at later time points may reflect altered differentiation dynamics result-ing from concurrent pharmacological stimulation and viral infection, rather than a simple additive effect. The relationship between precursor infection and PLP-associated E protein appears to follow a continuous biological association rather than a binary infected versus non-infected state.

Interestingly, our findings differ from those of Sridharan et al., [17] who reported that DENV infection suppressed megakaryocyte development in humanized mice. This discrepancy may reflect differences in models: in vivo bone marrow integrates stromal and immune interactions that can inhibit hematopoiesis, while our in vitro approach isolates virus–pre-cursor interactions and reveals an acceleration of differentiation. It is possible that in vivo, the combined effects of infec-tion and host responses produce net suppression, whereas in vitro, the direct viral effect is seen as premature maturation. Thus, the biological outcome of precursor infection may be context- and stage-dependent.

The presence of E protein in circulating platelets raises questions beyond platelet counts. Platelets are recognized as immune and inflammatory effectors [18]. E$^+$ platelets could interact abnormally with leukocytes, endothelium, or the coag-ulation system, potentially influencing disease manifestations such as vascular leakage or immune activation. Although our study was not designed to address platelet function, the detection of viral antigens in platelets suggests that these cells could serve as biomarkers of marrow precursor infection and possibly as mediators of pathophysiological processes in dengue.

Importantly, we do not claim a causal link between precursor infection and thrombocytopenia. While alterations in megakaryopoiesis have been reported in dengue, our data do not allow us to directly connect precursor infection with platelet counts in patients. Instead, we highlight that platelet carriage of viral proteins represents a measurable outcome of marrow infection, and one that correlates with clinical severity categories.

Limitations of our study include the use of immortalized cell lines (K562, MEG-01) rather than primary precursors, the modest number of patient samples, and the lack of functional assays of E$^+$ platelets. Expanding to larger cohorts and primary models will be needed to establish the clinical and mechanistic significance of platelet antigen carriage. Detailed

clinical metadata, including the precise timing of sample collection relative to symptom onset and quantitative viremia at the time of virus isolation, were not consistently available for all patients, which may influence the detection of viral antigens in platelets. Although viral replication kinetics were evaluated for selected reference strains and a representative clinical isolate (Fig 1), comprehensive kinetic analyses were not extended to all isolates examined in downstream differentiation assays; therefore, isolate-dependent differences in replication efficiency across the full panel cannot be completely excluded. In addition, absolute PLP numbers were not determined for all experiments, and variability in PLP recovery and cytometer acquisition required normalization to percentage values for reliable comparison across isolates. Finally, formal viability or apoptosis assays were not included, and thus potential contributions of cell stress or cell death to infection-associated differentiation and PLP production cannot be fully ruled out.

In conclusion, we provide evidence that DENV infection of megakaryocytic precursors is associated with accelerated differentiation and the generation of PLPs carrying dengue virus E protein. The observed correlation between in vitro infection of megakaryocytic precursors and the presence of E$^+$ platelets in dengue patients supports the concept that platelet-associated viral antigens may reflect infection at the level of bone marrow progenitors. While further studies will be required to define the functional consequences of E$^+$ platelets, our findings contribute to a better understanding of dengue pathogenesis and identify a potential cellular link between precursor infection and circulating platelet phenotypes.

## Acknowledgments

The authors would like to acknowledge Dr. Paola Carolina Valenzuela León from NIH for donating reagents, and all authors acknowledge the patients who voluntarily donated samples used in the present study.

## Author contributions

**Conceptualization:** Tannya Karen Castro Jiménez, Sergio Roberto Aguilar Ruiz, José Bustos Arriaga.

**Data curation:** Tannya Karen Castro Jiménez, Nallely Diaz Lima, José Alberto San Juan Luis, José Bustos Arriaga.

**Formal analysis:** Tannya Karen Castro Jiménez, José Alberto San Juan Luis, José Bustos Arriaga.

**Funding acquisition:** José Bustos Arriaga.

**Investigation:** Marcos Alvarado Silva, Tannya Karen Castro Jiménez, Edwin Antonio Lopez Kelly, Sergio Roberto Aguilar Ruiz, José Bustos Arriaga.

**Methodology:** Marcos Alvarado Silva, Tannya Karen Castro Jiménez, Edwin Antonio Lopez Kelly, Nallely Diaz Lima, José Alberto San Juan Luis, Diego Sait Cruz Hernández, José Bustos Arriaga.

**Project administration:** Tannya Karen Castro Jiménez, Nallely Diaz Lima, José Bustos Arriaga.

**Resources:** Nallely Diaz Lima, Diego Sait Cruz Hernández, José Bustos Arriaga.

**Software:** José Bustos Arriaga.

**Supervision:** Tannya Karen Castro Jiménez, José Bustos Arriaga.

**Validation:** Marcos Alvarado Silva, Tannya Karen Castro Jiménez, José Bustos Arriaga.

**Visualization:** José Bustos Arriaga.

**Writing – original draft:** Marcos Alvarado Silva, José Bustos Arriaga.

**Writing – review & editing:** Sergio Roberto Aguilar Ruiz, José Bustos Arriaga.

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
