## [Decision Letter · Decision Letter 0]

12 Nov 2025

Dear Dr. Bustos Arriaga,

Thank you for submitting your manuscript to PLOS ONE. After careful consideration, we feel that it has merit but does not fully meet PLOS ONE’s publication criteria as it currently stands. Therefore, we invite you to submit a revised version of the manuscript that addresses the points raised during the review process.

We look forward to receiving your revised manuscript.

Kind regards,

James P. Maloney, MD

Academic Editor

PLOS ONE

Additional Editor Comments:

This is an interesting manuscript regarding dengue pathogenesis. However both reviewers had substantial comments that need to be addressed in a major revision. Please note that one reviewer voted "reject" due to the large number of items that needed to be addressed, so the authors will need to carefully address all reviewer comments in a major revision. Failure to address these comments fully will likely lead to the rejection of a revised manuscript. As the authors may need to repeat experiments to address some of these comments, it is fair to allow 4 months for a major revision.

Reviewers' comments:

Reviewer's Responses to Questions

**Comments to the Author**

1. Is the manuscript technically sound, and do the data support the conclusions?

Reviewer #1: Partly

Reviewer #2: Partly

2. Has the statistical analysis been performed appropriately and rigorously?

Reviewer #1: No

Reviewer #2: Yes

3. Have the authors made all data underlying the findings in their manuscript fully available?

Reviewer #1: Yes

Reviewer #2: Yes

4. Is the manuscript presented in an intelligible fashion and written in standard English?

Reviewer #1: Yes

Reviewer #2: Yes

Reviewer #1: In this manuscript, “Evidence that intracellular dengue virus envelope protein in patient platelets originates from infection of megakaryocytic precursors,” the authors investigate whether dengue virus (DENV) infection of megakaryocytic precursor cells leads to the transfer of viral envelope (E) protein to platelets. They use K562 and MEG01 megakaryocytic cell lines, examine differentiation marker CD41, and detect DENV-E protein in platelet-like particles (PLPs) derived from infected cultures. Additionally, they report detection of DENV-E protein in platelets from dengue patients, with a positive correlation to disease severity.

The scientific question is important and clinically relevant. Understanding whether megakaryocytic precursors act as a reservoir for DENV proteins and contribute to platelet dysfunction has implications for dengue pathogenesis. The manuscript presents potentially interesting findings; however, several key aspects of experimental design, methodology, data presentation, and interpretation need to be improved. In its current form, the manuscript contains speculation that is not sufficiently supported by data, and many experiments lack methodological clarity.

Major Concerns

1-Insufficient methodological detail

Critical experimental parameters are either missing or unclear and needs to be addressed

• The infection protocol for K562 and MEG01 cells needs to be described in detail (MOI calculation, duration, infection conditions).

• It is unclear when PMA stimulation was performed relative to infection (pre-, post-, or simultaneous).

• The manuscript does not describe how infectivity of different virus isolates was determined before experiments.

• No explanation of how PLPs were distinguished from cell debris or apoptotic bodies during flow cytometry.

• Patient platelet isolation method requires a reference or procedural detail.

• Number of biological replicates per figure and statistical tests used are not indicated.

• multiple DENV isolates are compared, what normalization strategy was used should be defined. Different isolates may have different infectivity even at the same MOI; this must be addressed.

• Cell viability should have been included as DENV leads to apoptosis.

• Indicate number of biological replicates for each experiment.

• Include reference for patient platelet isolation protocol.

2-In results and figure section

1. Table 1 — Summarize patient demographics (n, sex distribution, severity categories).

2. Figure organization — Reorganize multi-panel figures (e.g., Fig. 1A–F as A with subpanels a–f).

3. Plaque assay terminology — Antibody-based quantification should be described as “focus-forming assay,” not plaque assay.

4. Figure 2 — Add a small legend or color explanation directly near the figure.

5. Figure 3 — Only performed in K562; please justify why MEG01 was not included.

6. Statistics — Explain why post-hoc analysis was used and ensure consistency across figures.

7. Figure 4 — Include additional methods to assess infection kinetics (viral RNA, infectious viral yield).

8. Figure 5 — Provide absolute PLP or platelet numbers, not just percentages.

9. Figure 9 — Group patients by severity rather than individual data points; include p-values. Clarify number of healthy controls.

Reviewer #2: Alvarado-Silva et al. have evaluated whether infection by various DENV strains, including patient isolates, of megakaryocyte precursors can transfer E protein to platelets. They propose E positive platelets may provide a marker of bone marrow infection. Many of the results are interesting, and this work is generally important for understanding dengue pathogenesis. However, there are numerous areas where the reviewer struggled to understand the figures and results as presented, detailed below.

Major comments:

• Line 74 says 10 blood samples were collected, but 11 patients are listed in table 1.

• Table 1: It would be helpful to add time since symptom onset when blood was collected as this may impact findings. Additionally, is viremia available for the patients from the time of virus isolation? This would also be helpful to include, if available.

• For Table 3, it would be helpful to include platelet counts, so it is possible to evaluate how severe the thrombocytopenia was.

• Line 158: how was it confirmed that the DENV E protein was intracellular and not E protein on the surface? Although intracellular staining was performed, this would stain both surface and intracellular protein.

• Line 236: at day 7 dpi, the CD41 MFI was only significantly higher in DENV1 than PMA, not than basal conditions. The way the sentence is written, it sounds like it was higher across both timepoints. Additionally, PMA + DENV is often lower (blue). It would be helpful if the authors helped explain this point.

• Figures 1 and 2: it would be helpful to understand why the authors evaluated the effect of the DENV1 reference strain on MEG-01 but not clinical isolates. It appears clinical isolates were only used on K562 cells.

• For all studies using clinical isolates, it was unclear what MOI was used. Could differences in viral concentrations have contributed to the variable differentiation?

• Figure 5: the methods say that Pearson correlation was performed, but the figure legend says Spearman’s correlation.

• For all figures where letters are provided instead of statistics comparing groups, it would be much easier of comparisons were presented as in Fig. 2, with bars and stars. It was difficult to follow which analyses were significant.

• It is very unclear what Fig. 10 is showing. Which in vitro infection was used for this comparison? It appears the authors are referring to the participant’s own clinical isolates, but this isn’t explained anywhere. Additionally, the legend does not seem to match the figure (labeled A and B in the legend, 3 panels in the actual figure) and all the axes have the same labels even though the data seem to be different in each figure.

• K562 cells were used as a model of megakaryocyte-erythroid progenitor cells. K562 cells are commonly used to evaluate antibody-dependent enhancement and are generally not considered infectable without the presence of DENV-specific serum to mediate enhancement. Is this why a high MOI had to be used to achieve infection? Providing some context about the more general use of K562 cells for studying dengue in either the introduction or discussion would be helpful.

Minor comments:

- Orthoflavivirus dengei – dengue?

- Fig. 6 and 7 F: label says 4dpi, should it say 7dpi?

- Please indicate what orange means in Fig. 9A and B. Dengue with Warning Signs?

**Do you want your identity to be public for this peer review?** For information about this choice, including consent withdrawal, please see our For information about this choice, including consent withdrawal, please see our Privacy Policy .

Reviewer #1: **Yes:** atoshi banerjeeatoshi banerjee

Reviewer #2: No

---

## [Author Response · Author response to Decision Letter 1]

16 Dec 2025

Response to the Academic Editor

We sincerely thank the Academic Editor for the careful assessment of our manuscript and for the opportunity to submit a major revision. We fully acknowledge the scope of the reviewers’ concerns and have addressed each comment in detail in the accompanying point-by-point response.

In this revised version, we substantially improved the manuscript by expanding methodological descriptions, clarifying experimental design, strengthening statistical analyses, and revising the presentation and interpretation of results to avoid overstatement. Figures have been reorganized and relabeled where needed, figure legends have been rewritten for clarity, and statistical reporting has been standardized across the manuscript.

For specific points where reviewers requested additional experiments, we carefully evaluated feasibility and have revised the manuscript to ensure that conclusions are fully supported by the available data. In cases where certain experiments could not be added at this stage due to limitations in the availability of specific clinical samples and/or key reagents, we explicitly acknowledged these limitations and clarified the scope of inference in the Discussion, adopting a conservative interpretative approach.

We believe that these extensive revisions address the reviewers’ concerns and significantly strengthen the manuscript. We respectfully submit the revised version for further consideration and appreciate the Editor’s guidance throughout this process.

Response to Reviewers

Manuscript ID: PONE-D-25-49675

Title: Evidence that intracellular dengue virus envelope protein in patient platelets originates from infection of megakaryocytic precursors

Authors: Alvarado-Silva et al.

Journal: PLOS ONE

We thank the Academic Editor and the reviewers for their careful evaluation of our manuscript and for the constructive comments provided. We have revised the manuscript extensively to address all concerns raised. Below, we provide a point-by-point response to each comment, with changes highlighted in the revised manuscript.

Reviewer #1: In this manuscript, “Evidence that intracellular dengue virus envelope protein in patient platelets originates from infection of megakaryocytic precursors,” the authors investigate whether dengue virus (DENV) infection of megakaryocytic precursor cells leads to the transfer of viral envelope (E) protein to platelets. They use K562 and MEG01 megakaryocytic cell lines, examine differentiation marker CD41, and detect DENV-E protein in platelet-like particles (PLPs) derived from infected cultures. Additionally, they report detection of DENV-E protein in platelets from dengue patients, with a positive correlation to disease severity.

The scientific question is important and clinically relevant. Understanding whether megakaryocytic precursors act as a reservoir for DENV proteins and contribute to platelet dysfunction has implications for dengue pathogenesis. The manuscript presents potentially interesting findings; however, several key aspects of experimental design, methodology, data presentation, and interpretation need to be improved. In its current form, the manuscript contains speculation that is not sufficiently supported by data, and many experiments lack methodological clarity.

We thank the reviewer for this careful assessment. We agree that greater methodological clarity was needed to ensure full transparency and reproducibility. In response, we have revised the Materials and methods section to explicitly describe infection conditions, stimulation timing, normalization strategies, flow cytometry gating, platelet isolation procedures, biological replicates, and statistical analyses. No new experiments were added; rather, previously implicit or dispersed methodological details were consolidated and clarified.

Major Concerns

1-Insufficient methodological detail

Critical experimental parameters are either missing or unclear and needs to be addressed

• The infection protocol for K562 and MEG01 cells needs to be described in detail (MOI calculation, duration, infection conditions).

We thank the reviewer for pointing this out. The infection protocol has now been explicitly detailed in the Materials and Methods section. We have clarified the multiplicity of infection (MOI) used for each cell line, the duration of viral adsorption, washing steps, and post-infection culture conditions to ensure reproducibility and transparency.

• It is unclear when PMA stimulation was performed relative to infection (pre-, post-, or simultaneous).

We agree that this point required clarification. The timing of PMA stimulation relative to viral infection has now been clearly specified. PMA was added simultaneously with viral infection and maintained throughout the culture period as a differentiation control, rather than as a pre-conditioning stimulus.

• The manuscript does not describe how infectivity of different virus isolates was determined before experiments.

We appreciate this observation. We have now clarified that viral infectivity was determined using an immunodetection-based focus-forming assay, which was used to quantify and normalize viral stocks prior to infection experiments. The terminology was also corrected to avoid confusion with classical plaque assays.

• No explanation of how PLPs were distinguished from cell debris or apoptotic bodies during flow cytometry.

We have now expanded the description of the flow cytometry gating strategy. PLPs were identified based on forward and side scatter properties relative to size calibration beads and by CD41 expression, while CD41-negative events and events outside the defined platelet/PLP gate were excluded to minimize inclusion of debris or apoptotic fragments.

• Patient platelet isolation method requires a reference or procedural detail.

The platelet isolation procedure has now been expanded and referenced to established protocols commonly used to minimize platelet activation during flow cytometric analysis. We also added reference 16.

• Number of biological replicates per figure and statistical tests used are not indicated.

We agree and have now clearly indicated the number of independent biological replicates and the statistical tests used for each analysis. This information has been added both to the Statistical Analysis section and to the corresponding figure legends.

• multiple DENV isolates are compared, what normalization strategy was used should be defined. Different isolates may have different infectivity even at the same MOI; this must be addressed.

We appreciate this important point. We have clarified that all viral isolates were normalized by infectious units and used at equivalent MOIs. While isolate-dependent differences in replication kinetics cannot be excluded, normalization by infectious input enabled comparative analysis across experiments. This limitation is now acknowledged in the text.

• Cell viability should have been included as DENV leads to apoptosis.

We acknowledge this limitation. Although formal viability or apoptosis assays were not included in this study, this has now been explicitly stated in the Discussion section. We have tempered our interpretations accordingly and identified this as an important aspect for future studies.

• Indicate number of biological replicates for each experiment.

We agree and have now clearly indicated the number of independent biological replicates and the statistical tests used for each analysis. This information has been added both to the Statistical Analysis section and to the corresponding figure legends.

• Include reference for patient platelet isolation protocol.

The platelet isolation procedure has now been expanded and referenced to established protocols commonly used to minimize platelet activation during flow cytometric analysis. We also added reference 15.

2-In results and figure section

1. Table 1 — Summarize patient demographics (n, sex distribution, severity categories).

We agree with the reviewer. Patient demographics were already included in Table 1; however, to improve clarity, a concise summary of the total number of patients, sex distribution, and clinical severity categories has now been added to the table legend.

2. Figure organization — Reorganize multi-panel figures (e.g., Fig. 1A–F as A with subpanels a–f).

We reorganized Figure 1 to clearly distinguish experimental conditions by using subpanels (a–f) within panel A, as suggested. Other figures already follow a standard quantitative multi-panel format (A–C, D–F) and therefore did not require subpanel restructuring

3. Plaque assay terminology — Antibody-based quantification should be described as “focus-forming assay,” not plaque assay.

We thank the reviewer for pointing this out. The quantification method used in the study was indeed an immunostaining-based assay, and “focus-forming assay (FFA)” is the correct terminology. All references to “plaque assay” have now been replaced with “focus-forming assay” in the Methods, Results, and figure legends for accuracy and consistency.

4. Figure 2 — Add a small legend or color explanation directly near the figure.

We thank the reviewer for this helpful suggestion. A color legend has now been added directly to Figure 2 (panels A and B) to clearly indicate the experimental conditions represented by each color (basal, PMA, DENV1, and PMA + DENV1).

5. Figure 3 — Only performed in K562; please justify why MEG01 was not included.

We thank the reviewer for this important comment. MEG-01 cells were initially included as a complementary megakaryocytic model at a more advanced differentiation stage, and preliminary experiments demonstrated that DENV infection induced changes in CD41 expression that were qualitatively comparable to those observed in K562 cells (Figure 2). However, MEG-01 cultures exhibited higher biological variability across time points and viral isolates, likely reflecting their more committed differentiation state and limited proliferative capacity.

To ensure experimental consistency and allow robust comparative analyses across multiple isolates and infection kinetics, subsequent experiments were therefore conducted using K562 cells as a well-established and stable model of megakaryocyte–erythroid progenitors. Importantly, K562 cells allowed reproducible assessment of isolate-dependent differences in differentiation and infection kinetics, which was the primary objective of Figure 3.

We have clarified this rationale in the revised manuscript.

6. Statistics — Explain why post-hoc analysis was used and ensure consistency across figures.

We thank the reviewer for highlighting the need to clarify the statistical approach. Post-hoc multiple-comparison tests were applied in experiments involving more than two experimental groups, such as comparisons among multiple viral isolates, treatment conditions (basal, PMA, DENV ± PMA), or time points. In these cases, an initial omnibus test was performed (one-way ANOVA or two-way repeated-measures ANOVA, as appropriate), followed by post-hoc testing to identify specific group-to-group differences while controlling for multiple comparisons.

The Methods section has now been revised to explicitly describe the statistical workflow used throughout the manuscript. Specifically, one-way ANOVA followed by Tukey’s post-hoc test was used for comparisons among multiple groups at individual time points; two-way repeated-measures ANOVA with Bonferroni correction was applied to replication-kinetics experiments; unpaired two-tailed Student’s t-tests were used when only two groups were compared; and correlation analyses were performed using Spearman’s rank correlation.

In addition, all figure legends have been carefully reviewed and updated to explicitly state the statistical tests applied, the post-hoc correction used when relevant, and the number of biological replicates.

7. Figure 4 — Include additional methods to assess infection kinetics (viral RNA, infectious viral yield).

We would like to clarify that infection kinetics were indeed assessed in this study and are presented in Figure 1. Specifically, viral replication kinetics in K562 and MEG-01 cells were evaluated by quantification of infectious virus released into culture supernatants using focus-forming assays. These analyses were performed using reference strains of DENV1, DENV4, and ZIKV, as well as one representative clinical isolate from Oaxaca.

The objective of Figure 4, however, was not to further characterize viral replication kinetics, but rather to examine the temporal effects of infection on megakaryocytic differentiation markers and platelet-like particle production following standardized infectious input. For this reason, viral RNA levels or infectious viral yield were not re-evaluated in Figure 4, and infections were normalized by infectious units to enable comparison of downstream cellular responses across conditions.

We have now clarified this distinction between figures in the Results section and have further discussed the scope and limitations of kinetic analyses in the Discussion.

8. Figure 5 — Provide absolute PLP or platelet numbers, not just percentages.

We thank the reviewer for this important observation. Assuming the comment refers to Figure 6, where PLP frequencies are quantified, we agree that presenting absolute PLP or platelet numbers can provide additional information. In our experiments, however, total PLP recovery and cytometer event acquisition varied substantially between replicates and isolates due to differences in culture density, differentiation efficiency, and flow cytometer sampling rate. For this reason, PLP values were normalized and expressed as percentages, which allowed consistent comparison across conditions.

Absolute PLP counts were not consistently available for all replicates and therefore could not be reliably incorporated. We have now clarified in the Results section and in the legend of Figure 6 why percentages were used, and we explicitly acknowledge this as a limitation in the Discussion.

9. Figure 9 — Group patients by severity rather than individual data points; include p-values. Clarify number of healthy controls.

We thank the reviewer for this important suggestion. In the revised version, Figure 9 has been updated to include healthy donors as an explicit comparison group and to display individual data points rather than bar graphs. This approach better reflects inter-individual variability and is more appropriate given the limited sample size.

Accordingly, statistical analyses were revised using non-parametric Mann–Whitney tests. While significant differences were consistently observed between dengue patients and healthy donors, differences between DWW and DWS/SD groups did not reach statistical significance, although a trend toward higher values in the more severe group remained.

The Results and Discussion sections have been revised to reflect these final analyses, and the figure panels presenting redundant bar representations were removed to improve clarity.

Reviewer #2: Alvarado-Silva et al. have evaluated whether infection by various DENV strains, including patient isolates, of megakaryocyte precursors can transfer E protein to platelets. They propose E positive platelets may provide a marker of bone marrow infection. Many of the results are interesting, and this work is generally important for understanding dengue pathogenesis. However, there are numerous areas where the reviewer struggled to understand the figures and results as presented, detailed below.

Major comments:

• Line 74 says 10 blood samples were collected, but 11 patients are listed in table 1.

We thank the reviewer for identifying this inconsistency. This was an error in the text: a total of 11 dengue patient samples were included in the study, as correctly shown in Table 1. Line 74 has been corrected to state that 11 samples were collected. We apologize for the oversight.

• Table 1: It would be helpful to add time since symptom onset when blood was collected as this may impact findings. Additionally, is viremia available for the patients from the time of virus isolation? This would also be helpful to include, if available.

In

---

## [Decision Letter · Decision Letter 1]

23 Feb 2026

Dear Dr. Bustos Arriaga:

Thank you for submitting your manuscript to PLOS ONE. After careful consideration, we feel that it has merit but does not fully meet PLOS ONE’s publication criteria as it currently stands. Therefore, we invite you to submit a revised version of the manuscript that addresses the points raised during the review process.

We look forward to receiving your revised manuscript.

Kind regards,

James P. Maloney, MD

Academic Editor

PLOS One

Journal Requirements:

Additional Editor Comments:

The revised manuscript is much improved. The original reviewer asked for a statistical analysis of the revision - so the authors will note a new statistical reviewer's comments that seem feasible to address.

Reviewers' comments:

Reviewer's Responses to Questions

**Comments to the Author**

Reviewer #1: All comments have been addressed

Reviewer #3: (No Response)

2. Is the manuscript technically sound, and do the data support the conclusions?

Reviewer #1: Yes

Reviewer #3: No

3. Has the statistical analysis been performed appropriately and rigorously?

Reviewer #1: I Don't Know

Reviewer #3: (No Response)

4. Have the authors made all data underlying the findings in their manuscript fully available?

Reviewer #1: Yes

Reviewer #3: Yes

5. Is the manuscript presented in an intelligible fashion and written in standard English?

Reviewer #1: Yes

Reviewer #3: Yes

Reviewer #1: The authors have addressed many of my previous comments and incorporated several recommended changes, resulting in improvement of the manuscript. The statistical analysis would benefit from review by a statistics expert, as I am not qualified to evaluate the appropriateness of the tests used. Overall, the manuscript is now improved and resolves many of the concerns raised in my initial review.

Reviewer #3: In this study the investigators investigated whether infection of megakaryocytic precursors contributes to the generation of platelets carrying dengue virus proteins. It appears that the statistical issues have been addressed, the one dealing with the analysis of multiple groups with post hoc comparisons has certainly been clarified. Reading the manuscript and track changes a number of clarifications and additions have been addressed as well. The revision is much improved. There are a few minor clarifications needed:

1. On Figure 5 what is the 45 degree line? Please explain that to the reader. Also the analysis looks like a method comparison approach. This is not necessarily handled with a correlation. One should consider a Bland Altman type of analysis as well.

2. On Figure 10 why the wide error bars on some of the points? Since this is apparently a regression the R squared is usually reported and not only the r as the authors have done.

3. Since the sample size is so small, then for some of the analyses, why not the non parametric Kruskal Wallis approach for a multiple group comparison and the Friedman test for repeated measures?

4. In Figure 10 since the two active groups are being compared to the control a Dunnet’s test might be considered.

**Do you want your identity to be public for this peer review?** For information about this choice, including consent withdrawal, please see our For information about this choice, including consent withdrawal, please see our Privacy Policy .

Reviewer #1: **Yes:** Atoshi BanerjeeAtoshi Banerjee

Reviewer #3: No

---

## [Author Response · Author response to Decision Letter 2]

2 Mar 2026

Response to Reviewers

Manuscript ID: PONE-D-25-49675

Title: Evidence that intracellular dengue virus envelope protein in patient platelets originates from infection of megakaryocytic precursors

Authors: Alvarado-Silva et al.

Journal: PLOS ONE

Reviewer #3: In this study the investigators investigated whether infection of megakaryocytic precursors contributes to the generation of platelets carrying dengue virus proteins. It appears that the statistical issues have been addressed, the one dealing with the analysis of multiple groups with post hoc comparisons has certainly been clarified. Reading the manuscript and track changes a number of clarifications and additions have been addressed as well. The revision is much improved. There are a few minor clarifications needed:

1. On Figure 5 what is the 45 degree line? Please explain that to the reader. Also the analysis looks like a method comparison approach. This is not necessarily handled with a correlation. One should consider a Bland Altman type of analysis as well.

Response:

We appreciate this important observation. The 45-degree reference line has been removed from Figure 5 in the revised manuscript to avoid any suggestion of a method-comparison or agreement analysis.

Figure 5 evaluates biological association between two distinct biological variables (percentage of infected cells and differentiation marker expression), not agreement between measurement methods. Therefore, a Bland–Altman analysis was not appropriate for this context.

We have clarified in both the figure legend and the Results section that each data point represents one independent infection condition (one viral isolate), plotted as the mean of biological replicates.

We have revised the figure and legend accordingly to clarify the nature of the analysis.

2. On Figure 10 why the wide error bars on some of the points? Since this is apparently a regression the R squared is usually reported and not only the r as the authors have done.

Response:

We thank the reviewer for this comment. In the revised version, error bars have been removed from Figure 10. The correlation analysis is now performed using one point per independent infection condition (mean of biological replicates), consistent with the experimental design.

Furthermore, we standardized the analysis using Spearman’s rank correlation coefficient. Because Spearman correlation does not model linear regression, R² is not applicable. Therefore, we report Spearman’s r and the corresponding two-tailed p value in the figure panels.

The Methods, Results, and figure legends have been revised accordingly to ensure clarity and consistency.

3:Since the sample size is so small, then for some of the analyses, why not the non parametric Kruskal Wallis approach for a multiple group comparison and the Friedman test for repeated measures?

Response:

We appreciate this suggestion. For correlation analyses (Figs 5 and 10), we now uniformly apply Spearman’s rank correlation, which does not assume normal distribution and is appropriate for small sample sizes.

For multi-group comparisons involving flow cytometry data, one-way ANOVA with Tukey’s post hoc test was retained because the data met parametric assumptions based on distribution assessment and homogeneity of variance within experimental replicates.

For replication kinetics experiments, two-way repeated-measures ANOVA was employed to model both viral strain and time as factors, including their interaction. The Friedman test does not accommodate factorial designs with interaction terms; therefore, two-way repeated-measures ANOVA was considered more appropriate for these experiments.

These statistical approaches are now explicitly justified in the revised Methods section.

4: In Figure 10 since the two active groups are being compared to the control a Dunnet’s test might be considered.

Response:

Figure 10 presents correlation analyses between two continuous variables (precursor infection and PLP-associated E protein levels) rather than group-wise comparisons against a single control group. Therefore, Dunnett’s multiple-comparison test is not applicable in this context.

This has been clarified in the revised figure legend and Results section.

Additional Clarifications

As part of this revision, we also:

Standardized correlation analyses in both Figures 5 and 10 using Spearman’s rank correlation.

Removed any reference to regression modeling or R².

Revised the Discussion to avoid overinterpretation of non-significant correlations at later time points.

Ensured consistency between statistical methods, figure legends, and Results descriptions.

We are grateful for the reviewer’s detailed statistical feedback, which has strengthened the rigor and clarity of our manuscript.

---

## [Editor Report · Decision Letter 2]

3 Mar 2026

Dear Dr. Bustos Arriaga,

We look forward to receiving your revised manuscript.

Kind regards,

James P. Maloney, MD

Academic Editor

PLOS One

Journal Requirements:

Additional Editor Comments:

The reviewer of the revised manuscript asked for a statistical review - that review is included below. The statistical "asks" can be dealt with in a minor revision.

Reviewers' comments:

In this study the investigators investigated whether infection of megakaryocytic precursors contributes to the generation of platelets carrying dengue virus proteins. It appears that the statistical issues have been addressed, the one dealing with the analysis of multiple groups with post hoc comparisons has certainly been clarified. Reading the manuscript and track changes a number of clarifications and additions have been addressed as well. The revision is much improved. There are a few minor clarifications needed:

1. On Figure 5 what is the 45 degree line? Please explain that to the reader. Also the analysis looks like a method comparison approach. This is not necessarily handled with a correlation. One should consider a Bland Altman type of analysis as well.

2. On Figure 10 why the wide error bars on some of the points? Since this is apparently a regression the R squared is usually reported and not only the r as the authors have done.

3. Since the sample size is so small, then for some of the analyses, why not the non parametric Kruskal Wallis approach for a multiple group comparison and the Friedman test for repeated measures?

4. In Figure 10 since the two active groups are being compared to the control a Dunnet’s test might be considered.

---

## [Author Response · Author response to Decision Letter 3]

5 Mar 2026

Response to Reviewers

Manuscript ID: PONE-D-25-49675

Title: Evidence that intracellular dengue virus envelope protein in patient platelets originates from infection of megakaryocytic precursors

Authors: Alvarado-Silva et al.

Journal: PLOS ONE

Reviewer #3: In this study the investigators investigated whether infection of megakaryocytic precursors contributes to the generation of platelets carrying dengue virus proteins. It appears that the statistical issues have been addressed, the one dealing with the analysis of multiple groups with post hoc comparisons has certainly been clarified. Reading the manuscript and track changes a number of clarifications and additions have been addressed as well. The revision is much improved. There are a few minor clarifications needed:

1. On Figure 5 what is the 45 degree line? Please explain that to the reader. Also the analysis looks like a method comparison approach. This is not necessarily handled with a correlation. One should consider a Bland Altman type of analysis as well.

Response:

We appreciate this important observation. The 45-degree reference line has been removed from Figure 5 in the revised manuscript to avoid any suggestion of a method-comparison or agreement analysis.

Figure 5 evaluates biological association between two distinct biological variables (percentage of infected cells and differentiation marker expression), not agreement between measurement methods. Therefore, a Bland–Altman analysis was not appropriate for this context.

We have clarified in both the figure legend and the Results section that each data point represents one independent infection condition (one viral isolate), plotted as the mean of biological replicates.

We have revised the figure and legend accordingly to clarify the nature of the analysis.

2. On Figure 10 why the wide error bars on some of the points? Since this is apparently a regression the R squared is usually reported and not only the r as the authors have done.

Response:

We thank the reviewer for this comment. In the revised version, error bars have been removed from Figure 10. The correlation analysis is now performed using one point per independent infection condition (mean of biological replicates), consistent with the experimental design.

Furthermore, we standardized the analysis using Spearman’s rank correlation coefficient. Because Spearman correlation does not model linear regression, R² is not applicable. Therefore, we report Spearman’s r and the corresponding two-tailed p value in the figure panels.

The Methods, Results, and figure legends have been revised accordingly to ensure clarity and consistency.

3:Since the sample size is so small, then for some of the analyses, why not the non parametric Kruskal Wallis approach for a multiple group comparison and the Friedman test for repeated measures?

Response:

We appreciate this suggestion. For correlation analyses (Figs 5 and 10), we now uniformly apply Spearman’s rank correlation, which does not assume normal distribution and is appropriate for small sample sizes.

For multi-group comparisons involving flow cytometry data, one-way ANOVA with Tukey’s post hoc test was retained because the data met parametric assumptions based on distribution assessment and homogeneity of variance within experimental replicates.

For replication kinetics experiments, two-way repeated-measures ANOVA was employed to model both viral strain and time as factors, including their interaction. The Friedman test does not accommodate factorial designs with interaction terms; therefore, two-way repeated-measures ANOVA was considered more appropriate for these experiments.

These statistical approaches are now explicitly justified in the revised Methods section.

4: In Figure 10 since the two active groups are being compared to the control a Dunnet’s test might be considered.

Response:

Figure 10 presents correlation analyses between two continuous variables (precursor infection and PLP-associated E protein levels) rather than group-wise comparisons against a single control group. Therefore, Dunnett’s multiple-comparison test is not applicable in this context.

This has been clarified in the revised figure legend and Results section.

Additional Clarifications

As part of this revision, we also:

Standardized correlation analyses in both Figures 5 and 10 using Spearman’s rank correlation.

Removed any reference to regression modeling or R².

Revised the Discussion to avoid overinterpretation of non-significant correlations at later time points.

Ensured consistency between statistical methods, figure legends, and Results descriptions.

We are grateful for the reviewer’s detailed statistical feedback, which has strengthened the rigor and clarity of our manuscript.

---

## [Editor Report · Decision Letter 3]

11 Mar 2026

Evidence that dengue virus envelope protein in patient platelets originates from infection of megakaryocytic precursors

PONE-D-25-49675R3

We’re pleased to inform you that your manuscript has been judged scientifically suitable for publication and will be formally accepted for publication once it meets all outstanding technical requirements.

Kind regards,

James P. Maloney, MD

Academic Editor

PLOS One

Additional Editor Comments: Thank your for answering the statistical reviewer's Comments

---

## [Editor Report · Acceptance letter]

PONE-D-25-49675R3

PLOS One

Dear Dr. Bustos Arriaga,

I'm pleased to inform you that your manuscript has been deemed suitable for publication in PLOS One. Congratulations! Your manuscript is now being handed over to our production team.

Kind regards,

on behalf of

Dr. James P. Maloney

Academic Editor

PLOS One